# Extrapolating Jet Radiation with Autoregressive Transformers

Anja Butter[1,2], François Charton[3], Javier Mariño Villadamigo[1],
Ayodele Ore[1], Tilman Plehn[1,4], and Jonas Spinner[1]

**1** Institut für Theoretische Physik, Universität Heidelberg, Germany
**2** LPNHE, Sorbonne Université, Université Paris Cité, CNRS/IN2P3, Paris, France
**3** Meta FAIR, CERMICS - Ecole des Ponts
**4** Interdisciplinary Center for Scientific Computing (IWR), Universität Heidelberg, Germany

October 21, 2025

## Abstract

Generative networks are an exciting tool for fast LHC event fixed number of particles. Autoregressive transformers allow us to generate events ~~with~~ containing variable numbers of particles,very much in line with the physics of QCD jet radiation, and offer the possibility to generalize to higher multiplicities. We show how ~~they~~ transformers can learn a factorized likelihood for jet radiation and extrapolate in terms of the number of generated jets. For this extrapolation, bootstrapping training data and training with modifications of the likelihood loss can be used.

# 1 Introduction

Modern LHC physics is defined by precision tests of the fundamental properties of particles and their interactions, both in and beyond the current Standard Model. The level of precision is continuously improved by experimental and theoretical progress, accompanied by but not limited to the rapidly increased LHC luminosity towards the high-luminosity LHC.

In view of this precision program, experimental and theoretical LHC physics are being transformed through modern machine learning (ML) [1, 2]. On the theory and simulation side, a range of neural network applications are improving every step of our first-principles simulation chain. This includes phase-space sampling [3–11], scattering amplitude surrogates [12–23], end-to-end event generation [24–28], and detector simulators trained on full simulations [29–51].

The main workhorses behind this transformation are generative networks. They learn phase space densities or Jacobians from simple distributions given sets of events and can reproduce these densities through fast sampling. Modern generative network architectures are normalizing flows, diffusion networks, and autoregressive transformers. Because they are fast, differentiable, and flexible, generative networks can enable new simulation and analysis strategies Theyse networks are reaching new levels of accuracy and can significantly amplify simulated training data [52, 53] and speed up the generation. Given the LHC requirements, they have to be controlled and precise in encoding kinematic patterns over an, essentially, interpretable phase space [54–58]. Conditional versions of the forward-generative networks allow for probabilistic unfolding [59–66] or inference through posterior sampling [67–69].

In this paper we tackle the physics problem of using generative networks to describe jets radiated from a hard scattering process. In fundamental QCD, jet radiation is described by successive probabilistic parton splittings. It is an integral part of QCD predictions for hadron colliders, where final states with a fixed number of jets are not in line with parton densities and collinear factorization [70–72]. The corresponding splitting kernels and the generated phase space correlations are approximately universal [73]. The generated number of jets follows well-defined patterns, also predicted by QCD.

Autoregressive generative networks can, just like with language, generate open-end sequences of particles, or events with a variable number of particles. An autoregressive generation requires a factorized phase space probability [74, 75]. This structure matches the QCD aspects of universal splittings and well-defined jet numbers. Our generative architecture of choice is an autoregressive transformer [28, 76]. An attractive benefit of this approach is the possibility of exploiting universal structures across jet multiplicities, which could allow for a single generative network to be deployed instead of a collection of specialized models. In this work we establish the fundamental idea by studying extrapolation to higher multiplicities, highlighting both the challenges and the opportunities.

The goal of this paper is to show, for the first time, that a generative transformer can extrapolate in the number of jets and generate approximately universal jet radiation for higher jet numbers than seen during the training. In Sec. 2 we describe the QCD structures motivating an approximately factorized phase space likelihood and its ML-realization, leading to our autoregressive generative transformer. We then present extrapolated predictions in the number of jets in Sec. 3, using bootstrapped training data in Sec. 3.2, a truncated loss without fixed stopping condition in Sec. 3.3, and a loss that overrides the stop condition in Sec. 3.4. In Appendix A we provide additional information on how to improve the accuracy of the generative transformer through including a classifier in the training, in the spirit of a GAN.

## 2 Autoregressive jet radiation

Given that jet radiation in QCD is described by universal splitting kernels and well-defined scalings in the number of jets, we will train an autoregressive transformer with a factorized likelihood loss to generate QCD jet radiation. The ultimate goal is to show that the transformer not only describes jet radiation to a number of jets represented in the training data, but that it can extrapolate to larger jet counts than seen during training.

We first remind ourselves of universal splittings in QCD and the typical scaling in the number of produced jets. We will then motivate our $Z$+jets dataset, exhibiting the universal so-called staircase scaling. To train a generative network we first derive a factorized phase space probability and then encode it in a loss function for an autoregressive transformer.

### 2.1 QCD jet radiation

Collinear parton splittings in the initial or final states are the backbone of QCD predictions for hadron colliders. Their universal nature is the basis of parton densities, parton showers, and jet radiation, and it defines the structure of LHC events [70–72]. A challenging consequence of collinear splittings is that any hard scattering process is accompanied by a variable number of jets in the final state, as described by jet radiation and parton showers in the multi-purpose event generators [77–80]. Combining parton shower and hard matrix element predictions is the theory basis for the entire precision physics program at the LHC [81–84].

**Universal autoregressive structure**

The physics background of our paper is the universal nature of jet radiation from collinear splittings, reflecting the collinear factorization of the matrix element and the phase space. It allows us to generate events with $n + 1$ final-state jets from events with $n$ final-state jets. For final state radiation this factorization is schematically written as

$$\sigma_{n+1} \sim \int \frac{dp^2}{p^2} dz \; \frac{\alpha_s}{2\pi} P(z)\sigma_n \,, \tag{1}$$

where $p^2$ is the invariant mass of the splitting parton, $z$ is the momentum fraction carried out of the hard process $\sigma_n$, and $P(z)$ are the universal collinear splitting kernels. In the initial state, this factorization is the basis of the DGLAP equation with the subtracted versions of the same collinear splitting kernels.

The iterative structure of Eq.(1) allows us to simulate parton splittings as Markov processes, and it also allows us to describe the underlying densities in an approximately factorized form. Such a factorized density is most efficiently generated by an autoregressive structure. The key ingredients are the perturbative QCD splitting functions and the non-splitting probability, referred to as Sudakov factor.

The actual simulation of, approximately, collinear jet radiation is not expected to be exact: first, we need to generate final transverse momenta for the radiated partons while keeping transverse momentum conservation [85]; second, we need to correct for color and spin correlations [86]; finally, the structure of successive $(1 \rightarrow 2)$-splittings might not be sufficient for the LHC precision [87, 88]. Nevertheless, the form of Eq.(1) suggests that in QCD events with increasing number of jets can be derived from a simple iterative pattern, and such a pattern can in principle be learned and extrapolated by a neural network with the right (autoregressive) architecture.

**Jet rate scaling**

The number of radiated jets in LHC events does not follow a universal distribution. However, we can derive two distinct patterns. Both are defined in terms of the ratio of $(n+1)$-jet to $n$-jet events or in terms of the fraction of events with $n$ jets,

$$R_{(n+1)/n} = \frac{\sigma_{n+1}}{\sigma_n} \qquad \text{and} \qquad P(n) = \frac{\sigma_n}{\sigma_{\text{tot}}} \qquad \text{with} \qquad \sigma_{\text{tot}} = \sum_{n=0}^{\infty} \sigma_n \, . \tag{2}$$

The ratios and the probabilities depend on kinematic cuts regularizing the soft and collinear divergences, typically the minimum transverse momentum of the counted jets, $p_{T,\text{min}}$.

1. The first pattern, Poisson scaling, implies in terms of the expectation value $\bar{n}$,

$$R_{(n+1)/n} = \frac{\bar{n}}{n+1} \qquad \Longleftrightarrow \qquad P(n) = \frac{\bar{n}^n e^{-\bar{n}}}{n!} \, . \tag{3}$$

   At colliders, it occurs for processes with large splitting probabilities and large scale differences, for instance multi-jet production in $e^+ e^-$ collisions.
2. We focus on the alternative staircase scaling [89–91] with

$$R_{(n+1)/n} = e^{-b} \qquad \Longleftrightarrow \qquad P(n \geq n_{\text{min}}) = e^{-b n_{\text{min}}} \, . \tag{4}$$

   While the ratio $e^{-b}$ is the same for the exclusive and inclusive jet counts, the probability only has a simple form for the inclusive jet count, classifying events with $n_{\text{min}}$ jets or more. We can use the universal scaling to relate $P(n)$ to a successive or conditional probability

$$P(n+1|n) = R_{(n+1)/n} \, . \tag{5}$$

At colliders, staircase scaling is predicted for smaller splitting probabilities and democratic scales [92]. In that case, the jet count distributions can be derived from QCD using generating functionals [93]. For final state radiation we quote the scale-dependent result

$$R_{(n+1)/n} = 1 - \tilde{\Delta}_g(Q^2) \, , \tag{6}$$

with a modified Sudakov factor or non-splitting probability

$$\tilde{\Delta}_g(Q^2) = \exp\left[ -C_A \int_{Q_0^2}^{Q^2} dt \, \frac{\alpha_s(t)}{2\pi t} \left( \log \frac{t}{Q_0^2} - \frac{11}{6} \right) \right] \, . \tag{7}$$

To leading-log level the integrand is the QCD splitting function in the collinear approximation. This QCD derivation of staircase scaling requires democratic scales $Q^2/Q_0^2 \sim \mathcal{O}(1)$.

At the LHC the standard example is weak boson production with jets,

$$pp \to Z + n \, \text{jets} \qquad \text{with} \qquad n = \{0, 1, 2, 3, ...\} \tag{8}$$

Because the two scaling patterns are different, we will limit ourselves to learning and generating staircase scaling from datasets described by universal collinear radiation.

## 2.2 *Z + jets dataset*

We follow the above motivation and Refs. [28,54] by generating $Z$ bosons ~~leptonically~~ decaying to muons in association with a variable number of jets. Unlike for earlier studies, we include higher jet multiplicities to provide a challenge for the transformer

$$pp \to Z_{\mu\mu} + \{0, \cdots, 10\} \, \text{jets}. \tag{9}$$

We use MADGRAPH5_AMC_@NLOv3.5.1 [94] to generate 500M $pp$ events at a center-of-mass energy of $\sqrt{s} = 13$ TeV, including ISR and parton shower with PYTHIA8 [78], using CKKW merging [81] and hadronization, but no pile-up. The jets are defined with FASTJETv3.3.4 [95] using the anti-$k_T$ algorithm [96] with $R = 0.4$ and the basic cuts

$$p_{T,j} > 20 \text{ GeV} \qquad \text{and} \qquad \Delta R_{jj} > 0.4. \tag{10}$$

The muons and jets are both ordered in descending transverse momentum. Our phase space dimensionality is three per muon and four per jet. Momentum conservation is not guaranteed, because some final-state particles might escape for instance the jet algorithm. The distribution of the number of jets and the corresponding ratios $R_{(n+1)/n}$ are shown in the two panels of Fig. 1. We observe an approximately constant ratio for most of the spectrum, confirming a staircase scaling as defined in Eq.(4). Towards large numbers of jets we start encountering statistical limitations as well as phase space limitations.

Of our 500M events we use 80% for training, 10% for validation, and 10% for testing. The number of events per jet multiplicity is given in Tab. 1. To avoid being entirely dominated by low-multiplicity events, we cap the number of events with $n = 0, 1, 2$ to match the number of events with $n = 3$.

For the jet momenta, we use a minimal preprocessing [28, 54], where each particle $i$ is represented in standard jet coordinates

$$\{ (p_T, \eta, \phi, m)_i \} . \tag{11}$$

We enforce the $p_T$ cuts in Eq.(10) using the transformation $\log(p_T - p_{T,\min})$, which maps allowed transverse momenta to the full real line and leads to an approximately Gaussian shape. The jet mass is encoded as $\log m$. We express angles $\phi$ relative to the leading muon and apply a special treatment described in Section 2.4 to reflect the periodicity. We suppress the leading muon $\phi$ angle due to the global rotation symmetry. Finally, we standardize all phase space variables except $\phi$ as $(x - \bar{x})/\sigma(x)$. For 10 jets ~~the~~ this phase space is 45-dimensional.

## 2.3 Factorized probability

Following the discussion in Sec. 2.1, QCD jet radiation has two features that make it an attractive target for autoregressive generative networks: the universal splitting kernels and the

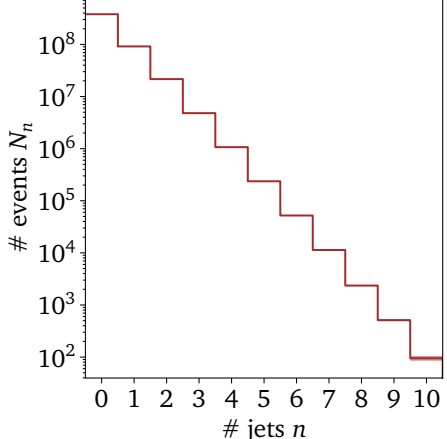
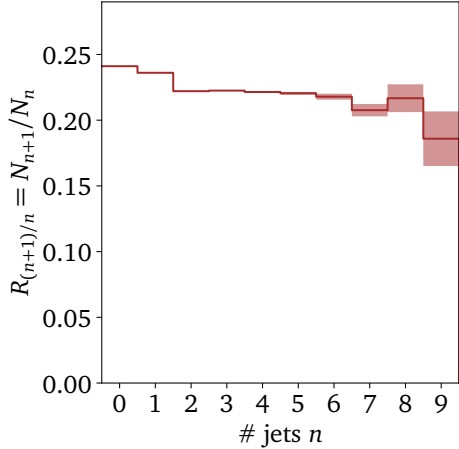

Figure 1: Staircase scaling of the number of jets in our $pp \to Z + n$ jets dataset. We show statistical uncertainties and use Gaussian error propagation to estimate the uncertainties for the ratio $R_{n+1/n}$.

| Number of jets | 0 | 1 | 2 | 3 | 4 | 5 | 6 | 7 | 8 | 9 | 10 |
|---|---|---|---|---|---|---|---|---|---|---|---|
| Number of events | 380M | 91M | 21M | 4.7M | 1.1M | 230k | 52k | 11k | 2.3k | 510 | 95 |
| Cap | 4.7M | 4.7M | 4.7M | - | - | - | - | - | - | - | - |

Table 1: Event counts for our simulated $Z$+jets dataset. When training networks, we cap the size of the 0,1,2-jet subsets.

jet ratio patterns. In case of staircase scaling the ratios of exclusive and inclusive jet rates are also universal. Equation (1) suggests that the phase space density for an event $x$ can be constructed as a product of conditional distributions, each taking the form

$$p(x_i|x_{1:i-1}) = p_{\text{kin}}(x_i|x_{1:i-1})\, p_{\text{split}}(x_{1:i-1})\,, \tag{12}$$

where we denote by $x_{1:i-1}$ the sequence of ~~particles~~progenitor partons $x_1, \ldots, x_{i-1}$. For particle $i$, $p_{\text{kin}}$ encodes the kinematics, conditional on the probability $p_{\text{split}}$ that it will be radiated. Both probabilities are conditioned on the full previous sequence of particles $x_{1:i-1}$. This dependence on the complete previous sequence is necessary to describe non-Markovian corrections to the universal nature of splitting kernels.

Approximate universality of the splitting kernels and jet ratios translates to universality of $p_{\text{kin}}$ and $p_{\text{split}}$ respectively. This raises the possibility that, given the right architecture, we can train a neural network to extrapolate QCD jet radiation patterns in analogy to a collinear parton shower Monte Carlo approach.

Using the conditional probabilities in Eq.(12) we can build the likelihood of an $n$-jet event,

$$
\begin{aligned}
p(x_{1:n}) &= \left[\prod_{i=1}^{n} p(x_i|x_{1:i-1})\right] \left[1 - p_{\text{split}}(x_{1:n})\right] \\
&= \left[\prod_{i=1}^{n} p_{\text{kin}}(x_i|x_{1:i-1})\right]\left[\prod_{i=1}^{n} p_{\text{split}}(x_{1:i-1})\right]\left[1 - p_{\text{split}}(x_{1:n})\right],
\end{aligned}
\tag{13}
$$

where the last term gives the probability that there are no further splittings and the event is complete. In QCD language it corresponds to a Sudakov factor. In accordance with Eq.(11), the phase space probability $p(x_{1:n})$ has a well-defined dimensionality $4^n$. It is normalized both as a continuous distribution over $x_i$ and a categorical distribution over $n$,

$$\sum_{n=1}^{\infty} \int \mathrm{d}x_1 \ldots \mathrm{d}x_n\, p(x_{1:n}) = 1\,. \tag{14}$$

As illustrated in Fig. 2, the generative process can be visualized as a binary probability tree with a Sudakov stop if no further splitting happens. The combination of $p_{\text{split}}$ for a splitting or $(1 - p_{\text{split}})$ for no splitting is described by a Bernoulli distribution $p_{\text{bin}}$ with expected splitting probability $p_{\text{split}}$

$$p_{\text{bin}}\left(y|p_{\text{split}}\right) = p_{\text{split}}^{y}(1 - p_{\text{split}})^{1-y} \quad \text{with} \quad y \in \{0, 1\},\, p_{\text{split}} \in [0, 1]\,. \tag{15}$$

It allows us to unify the factors $p_{\text{split}}$ and $1 - p_{\text{split}}$ in a completely factorized likelihood

$$p(x_{1:n}) = \prod_{i=1}^{n} p_{\text{kin}}(x_i|x_{1:i-1}) \prod_{i=0}^{n} p_{\text{bin}}\left(1 - \delta_{in}|p_{\text{split}}(x_{1:i})\right)\,. \tag{16}$$

The Kronecker delta $\delta_{in}$ assigns the splitting label zero for the $n^{\text{th}}$ particle and one otherwise. By keeping the full conditioning on $x_{1:i}$, this likelihood is completely general and can capture

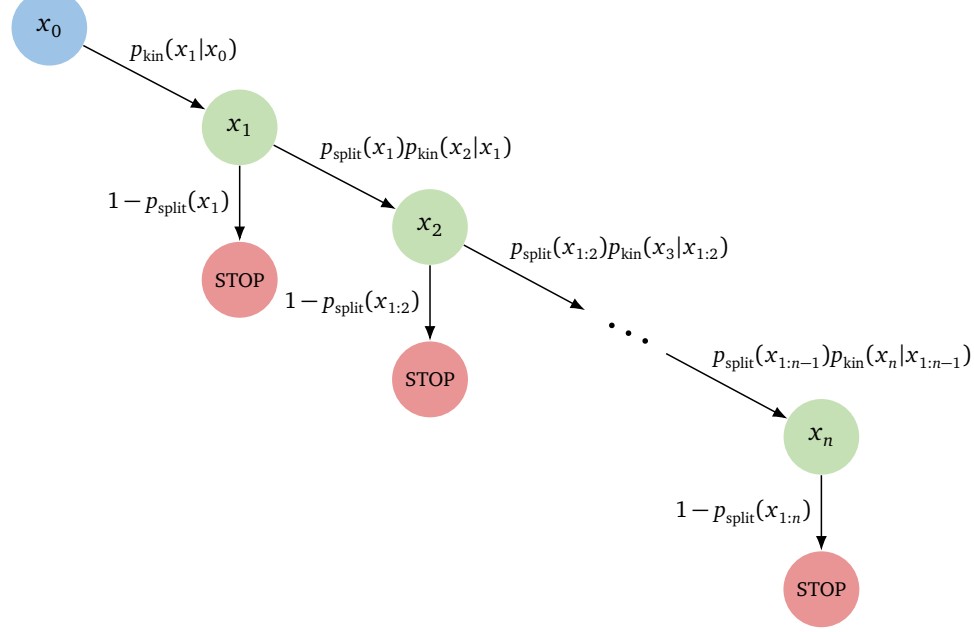

Figure 2: Probability tree for variable-length event generation. To disallow empty events, we assign $p_{\text{split}}(x_0) = 1$.

non-universal correlations. This is important when we describe full events, including the hard process. For $Z_{\mu\mu}$+jets events, we also ~~treat the muons autoregressively and enforce a splitting probability of one for them~~ include the muons in the sequence, but explicitly set their splitting probabilities to one instead of learning them. In addition, we use an additional one-hot encoded network input to distinguish the two types of muons and the jets.

Similarly to the ~~autoregressive~~ decomposition of the ~~likelihood of different particles~~ event likelihood $p(x_{1:n})$, we autoregressively factorize the likelihood of individual particles $p_{\text{kin}}(x_{i+1}|x_{1:i})$ in terms of their components. The ordering of components can affect the network performance [28], however for such small sequences this effect is negligible. The elements of the sequence are one-dimensional, and we parametrize their distributions with mixtures

$$\begin{aligned}
p_{\text{kin}}(x_{i+1}|x_{1:i}) = &\; p_{\text{GM}}(p_{T,i+1}|x_{1:i}) p_{\text{vMM}}(\phi_{i+1}|x_{1:i}, p_{T,i+1}) \\
&\times p_{\text{GM}}(\eta_{i+1}|x_{1:i}, p_{T,i+1}, \phi_{i+1}) p_{\text{GM}}(m_{i+1}|x_{1:i}, p_{T,i+1}, \phi_{i+1}, \eta_{i+1}).
\end{aligned} \tag{17}$$

We use Gaussian mixtures $p_{\text{GM}}$ for non-periodic variables and von Mises mixtures $p_{\text{vMM}}$ for the periodic variable $\phi$. Since it is straightforward to sample and to compute densities with these component-wise mixtures, the same is true for the full event likelihood. We do not generate the fixed muon mass in $Z_{\mu\mu}$+jets events. Periodic likelihoods for angular variables inform the network about this geometric information and therefore improve the performance. This has been previously shown for normalizing flows [97] and conditional flow matching [19].

In contrast to the autoregressive structure of $p(x_{1:n})$ in Eq.(13), Eq.(17) is not inspired by physics and other choices are possible. Examples from the literature are categorical distributions over bins (which suffer from limited resolution) [28,76,98], normalizing flows [69,99], and conditional flow matching [69,99].

We emphasize that this factorized likelihood, built to describe an autoregressive generation, generalizes the usual factorization $p(x_{1:n}) = p(x_{1:i}|i)p(i)$ [54,69] and previous autoregressive approaches [28]. Similar generative approaches have been developed for jet constituent generation [100], and a similar factorization for density estimation has been studied in Refs. [74,75].

## 2.4 Autoregressive transformer

Starting from the physics-motivated factorization in Eq.(16), we need to encode these densities with variable-length inputs $x_{1:i}$ using neural networks. A transformer $f_\theta$ with causal attention mask will turn these sequences into fixed-sized representations. We use a pre-layernorm transformer decoder with GeLU activations, for more information see App. B, and decompose the transformer output as

$$f_\theta(x_{1:i}) = (\rho_i, v_i) \in \mathbb{R} \times \mathbb{R}^d \,. \tag{18}$$

The embedding dimension $d$ is a hyperparameter. The $\rho_i$ represent the splitting probabilities that parametrize the Bernoulli distributions,

$$\rho_i \approx p_{\text{split}}(x_{1:i}) \,. \tag{19}$$

The embeddings $v_i$ similarly parametrize the kinematic conditionals

$$p_{\text{kin}}(x_i|v_{i-1}) \approx p_{\text{kin}}(x_i|x_{1:i-1}) \,. \tag{20}$$

For clarity, we always suppress the dependence of $\rho_i$ and $v_i$ on $x_{1:i}$ and on the transformer parameters $\theta$.

**Loss and training**

The loss function of the autoregressive network is given by the likelihood in Eq.(16),

$$\begin{aligned}
\mathcal{L}_{\text{like}} &= -\Big\langle \log p(x_{1:n}) \Big\rangle_{x \sim p_{\text{data}}} \\
&= -\Big\langle \sum_{i=1}^{n} \log p_{\text{kin}}(x_i|v_{i-1}) + \sum_{i=0}^{n} \log p_{\text{bin}}(1-\delta_{in}; \rho_i) \Big\rangle_{x \sim p_{\text{data}}} \,.
\end{aligned} \tag{21}$$

The first term is the usual likelihood loss for the kinematic generative network. The second term is the standard binary cross entropy. In our generative network it implicitly enforces the correct event multiplicity through a splitting discriminator.

In Sec. 3 we will consider modified training strategies to extrapolate beyond the maximal multiplicity $n_{\text{max}}$ of events contained in the training dataset. One strategy is to modify the cross entropy part of the likelihood loss in Eq.(21), for example by removing the contribution from the term with highest multiplicity $n_{\text{max}}$

$$\mathcal{L}_{\text{trunc}} = \Big\langle -\sum_{i=1}^{n} \log p_{\text{kin}}(x_i|v_{i-1}) - \sum_{i=0}^{n} (1-\delta_{in_{\text{max}}}) \log p_{\text{bin}}(1-\delta_{in}; \rho_i) \Big\rangle_{x \sim p_{\text{data}}} \,. \tag{22}$$

Using this loss, the splitting prediction for maximum-multiplicity events, $\rho_{n_{\text{max}}}$, is not explicitly trained. Rather, the weight sharing in the transformer allows correlations learned at lower multiplicity to be recycled.

When training our transformers on the $Z$+jets dataset from Sec. 2.2, we use the Adam optimizer with constant learning rate $3 \times 10^{-4}$ and batch size 1024. The batches contain events with different multiplicities following the distribution in the training data. The validation loss is tracked every 5k iterations, and we restore the network from the checkpoint with lowest validation loss after 200k iterations.

**Sampling**

To generate full events $x$, we sequentially sample from the likelihood described in Sec. 2.3, as visualized in Fig. 2. We sample 10M events in total and split them according to their multiplicities. This procedure generates samples from the exact likelihood learned by the network, but does not give us explicit control over the generated jet multiplicities. We decide on a maximum number of jets, and discard events for which the transformer predicts further splittings.

**Bayesian network**

Because we hope to use the autoregressive transformer for extrapolation beyond the jets present in the training data, we need to quantify the uncertainty in the predicted phase space density. We resort to Bayesian neural networks (BNN) [101–104] as a way to learn systematic and statistical uncertainties together with the mean network predictions. These are a standard method in LHC physics, for instance for amplitude regression [17], calibration [105,106], and classification [107]. They can be generalized to the density estimation aspect of generative networks [28,54,108,109], where they return an uncertainty on the unit event weight.

BNNs replace the network parameters $\theta$ by learnable distributions $q(\theta)$, usually assumed to be uncorrelated Gaussians. Their loss consists of a sampled likelihood term and a regularization with a prior-width hyperparameter,

$$\mathcal{L}_{\text{BNN}} = -\Big\langle \log p(x) \Big\rangle_{x \sim p_{\text{data}}, \theta \sim q} + D_{\text{KL}}\big[q(\theta), p(\theta)\big] \ . \tag{23}$$

To evaluate the BNN we sample from the learned weight distributions, in our case generating 10 samples, with a given number of 10M events each.

## 3 Results

Even though we are interested in extrapolating towards unseen jet numbers, we first benchmark the accuracy of our transformer in Sec. 3.1. We also show how without modifications the generative network does not actually extrapolate. For a successful extrapolation we first use a bootstrap approach in Sec. 3.2 and then show in Sec. 3.3 and Sec. 3.4 how truncating or overriding the likelihood loss allows the network to generate larger jet numbers than seen during training.

### 3.1 Generating without extrapolation

We begin by demonstrating that our transformer learns the phase space density precisely across event multiplicity. We train a Bayesian version of the transformer using all $Z + n$ jet events, from the hard process only, or $n = 0$, up to $n = 10$. We sample 10M events each from 10 BNN predictions. The jet multiplicity distribution is shown in the left panel of Fig. 3, showcasing that the generator can learn the universal staircase scaling. In Fig. 4, we see that the network reproduces the kinematic distributions with precision down to the statistical uncertainty of the test set. The transverse component of the vector sum of all particle momenta, $p_{T,\Sigma}$, provides a sensitive test of learned global correlation among all particles. All deviations from the training data are captured by the Bayesian uncertainty.

Next, we inspect whether the network has learned universal structure in the probability to generate additional jets. In the right panel of Fig. 3 we show the distributions of $p_{\text{split}}$ predicted during the autoregressive sampling steps. We train the network on the entire dataset, with up

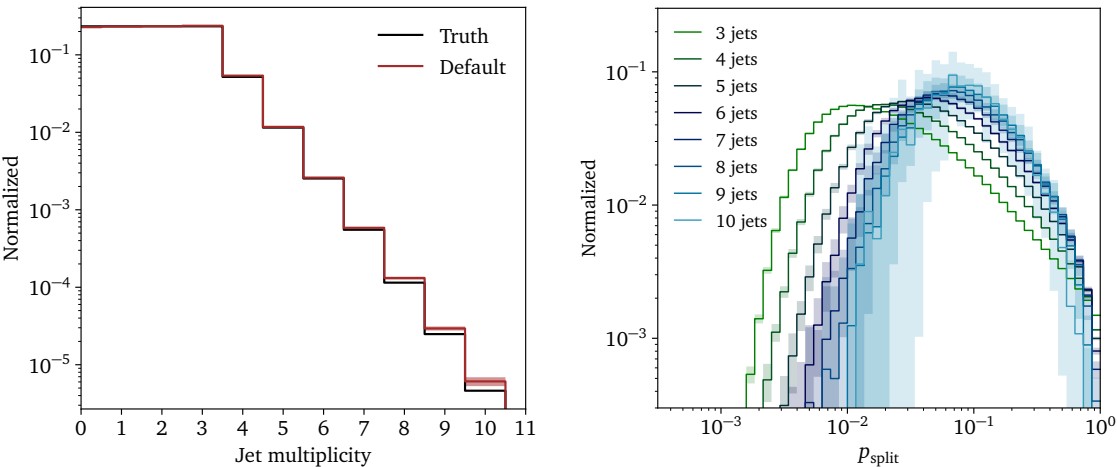

Figure 3: Jet multiplicity distribution (left) and splitting probabilities (right) for samples generated with the transformer trained on the full dataset up to 10 jet events.

to 10 jets. We ignore the learned $p_{\text{split}}$ for the first two jets, because we manually capped the number of training events for up to two jets, as shown in Tab. 1. For more than 6 jets, the distribution stabilizes within the Bayesian uncertainty band, indicating that between 6 and 10 jets we do not observe a significant effect from the parton densities [93].

**Naive extrapolation**

Because the termination of the number of jets is implemented probabilistically, we can naively extrapolate to higher jet numbers. For instance, we can train the networks with up to 6 jets and

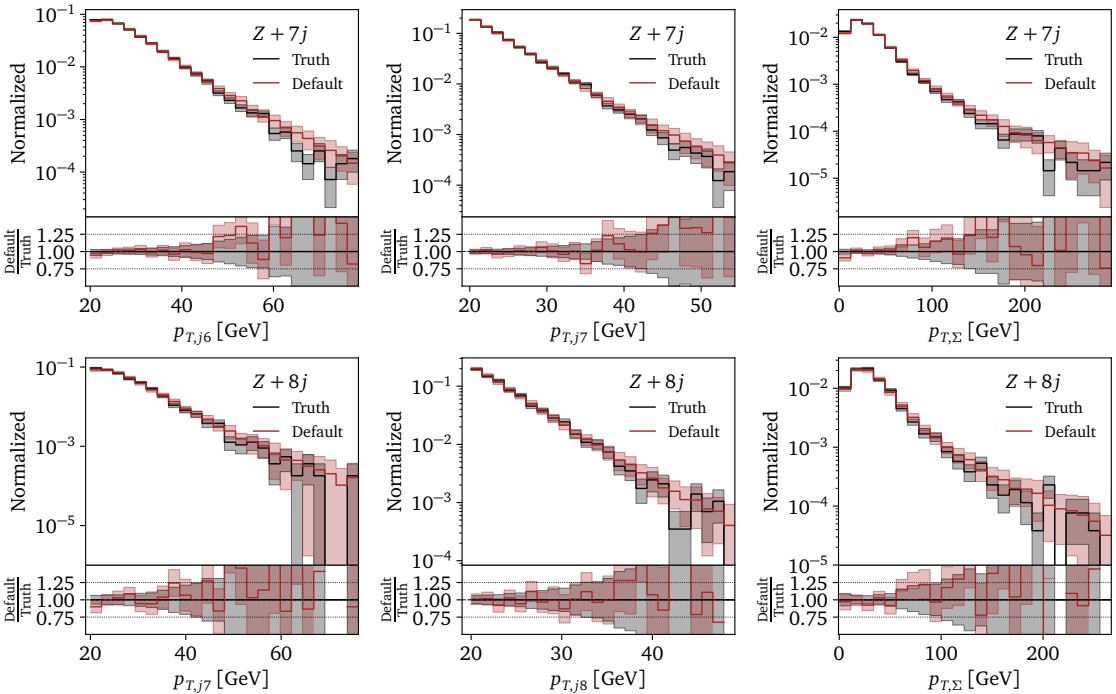

Figure 4: Selection of features in $Z + 7$ and 8-jet events for the generative network trained on the full dataset, including 7 and 8-jet events.

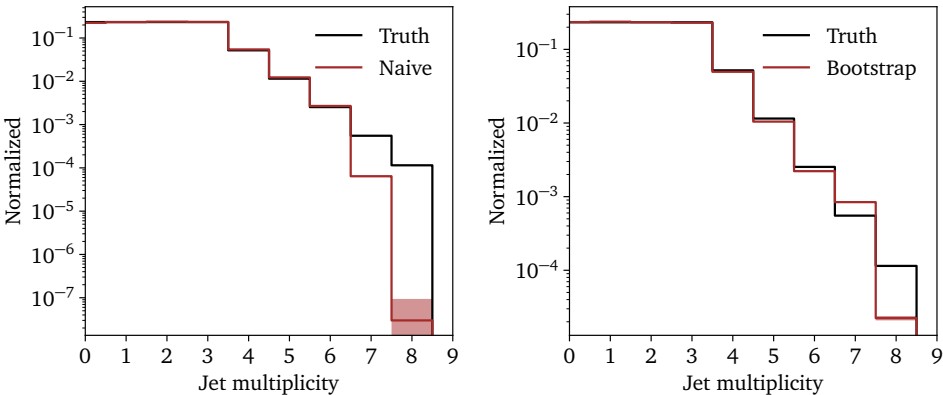

Figure 5: Jet multiplicity distributions for samples generated with the transformer using the naive training (left) and bootstrapping (right).

assess the small number of 7-jet and 8-jet events they generate. While the quality of 7-jet and 8-jet events should be worse than for jet numbers seen during training, we want to know if the transformer can leverage universal properties of the QCD jet radiation. We show the generated jet multiplicity distribution in the left panel of Fig. 5. Indeed, the network generates events with more than 6 jets, albeit with much lower probability than expected from staircase scaling.

For perfect training, we expect the rate for events with more jets than the training set to approach zero. This is because the transformer output $\rho_i$ is trained to match the probability that another jet follows particle $i$, $\rho_i \approx p_{\text{split}}(x_{1:i})$. In a given training set, with maximum event length $n_{\text{max}}$, one always has

$$p_{\text{split}}(x_{1:n_{\text{max}}}) = 0 \,. \tag{24}$$

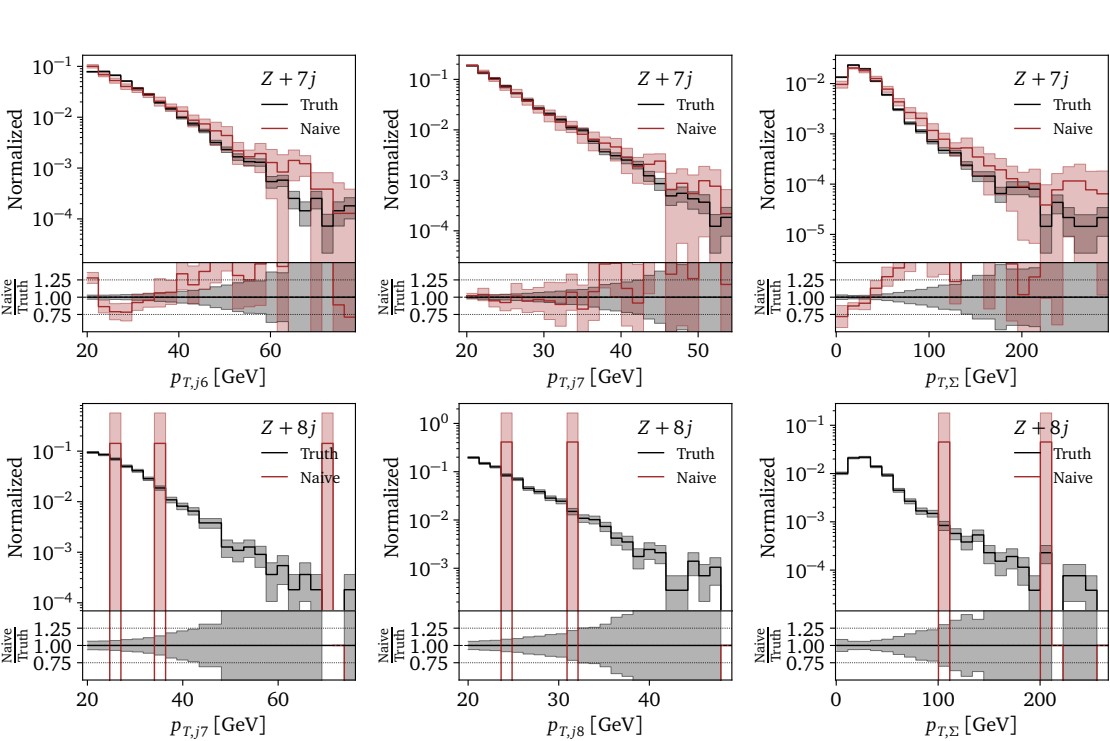

Figure 6: Selection of features in $Z + 7$ and 8-jet events for a generator trained on up to 6-jet events.

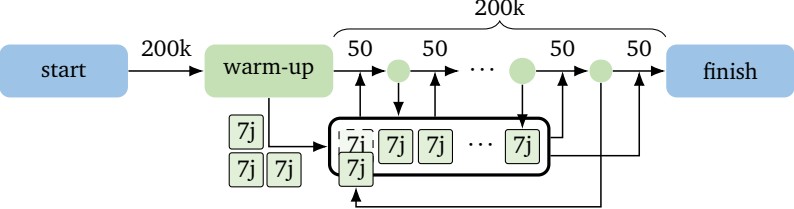

Figure 7: Training workflow for extrapolation with bootstrap. The upper horizontal arrows denote iterations within the training process, and vertical arrows denote moving bootstrapped events to update the buffer and include buffered events in training. Additional information is given in the text.

The optimal network would learn $\rho_{n_{\max}} = 0$, and the transformer can ignore physical correlations. The reason we do not observe exact zero splitting probabilities is that the weight sharing in the autoregressive transformer imparts a bias to reuse the pattern learned at low multiplicities.

Given the small but finite rate of 7-jet events generated through naive extrapolation, we want to see if the transformer has generalized the jet kinematics. In Fig. 6, we show the kinematic features as for the extrapolated 7-jet events. Among 100M generated events, the transformer only generates three 8-jet events, so we cannot assess their quality. However, the 7-jet events look qualitatively reasonable. In particular, the slightly broken transverse momentum conservation is reproduced with an accuracy similar to the baseline in Fig. 4. Given that the $p_T$ of the $7^{\text{th}}$ jet is approximately the same scale as the level of momentum non-conservation, this is a non-trivial result. It suggests that the transformer indeed generalizes kinematics, and we should mainly address the learned jet multiplicity.

### 3.2 Extrapolation with bootstrap

A simple modification to increase the fraction of learned 7-jet events is to bootstrap them, i.e. add generated 7-jet events to the training data. This way, we dynamically break the condition $p_{\text{split}}(x_{1:n_{\max}}) = 0$ of Eq.(24) and allow the network to adapt its multiplicity distribution. By repeating this bootstrapping, we can also generate 8 jets and beyond. The fraction of generated events introduced to the training dataset is a hyperparameter. It controls the learned multiplicity distribution.

The training workflow is visualized in Fig. 7. We start to add bootstrapped events after a warm-up stage of 200k iterations, corresponding to a full-length training in the approach of Sec. 3.1. Without this warm-up stage, the network memorizes the poor-quality samples of the freshly initialized network. After the warm-up, we generate a buffer of 1k 7-jet events. For every generated batch we sample a new deterministic network from the learned weight distribution, making sure that we cover the full range of the weight posterior distribution. We then add a single 7-jet event to each batch of 1024 events, corresponding roughly to the fraction of 7-jet events in the training dataset, and train for another 200k iterations with these settings. After every 50 iterations, we generate a batch of 32768 events, extract the 7-jet events and add them to the buffer. Once the buffer contains 50k events, we start to replace its oldest events with newly generated events. This allows the network to dynamically adapt the quality of 7-jet events. We observe that the network has to be trained for a sufficient amount of time in the bootstrapping mode to adapt to the changed multiplicity distribution.

The obtained jet multiplicity distribution is shown in the right panel of Fig. 5. We now get significantly more 7-jet and 8-jet events, indicating that the network indeed adapts the

multiplicity distribution. The fraction of 8 jet events is significantly lower than in the training data, because we only bootstrap 7 jet events. The kinematical distributions of the generated events are shown in Fig. 8. They show that the bootstrapping generator yields valid kinematic configurations. However, there are deviations in the kinematic features from the truth that are not covered by the Bayesian uncertainty. We remark, however, that Bayesian uncertainties are not expected to cover out-of-distribution deviations.

## 3.3   Extrapolation with truncated loss

A complementary way to combat the suppression of events with more jets than the training set is to modify the likelihood loss. As discussed in Sec. 3.1, the cause of the suppression is the constant $p_{\text{split}}(x_{1:n_{\max}}) = 0$ represented by a training dataset with at most $n_{\max}$ jets. A

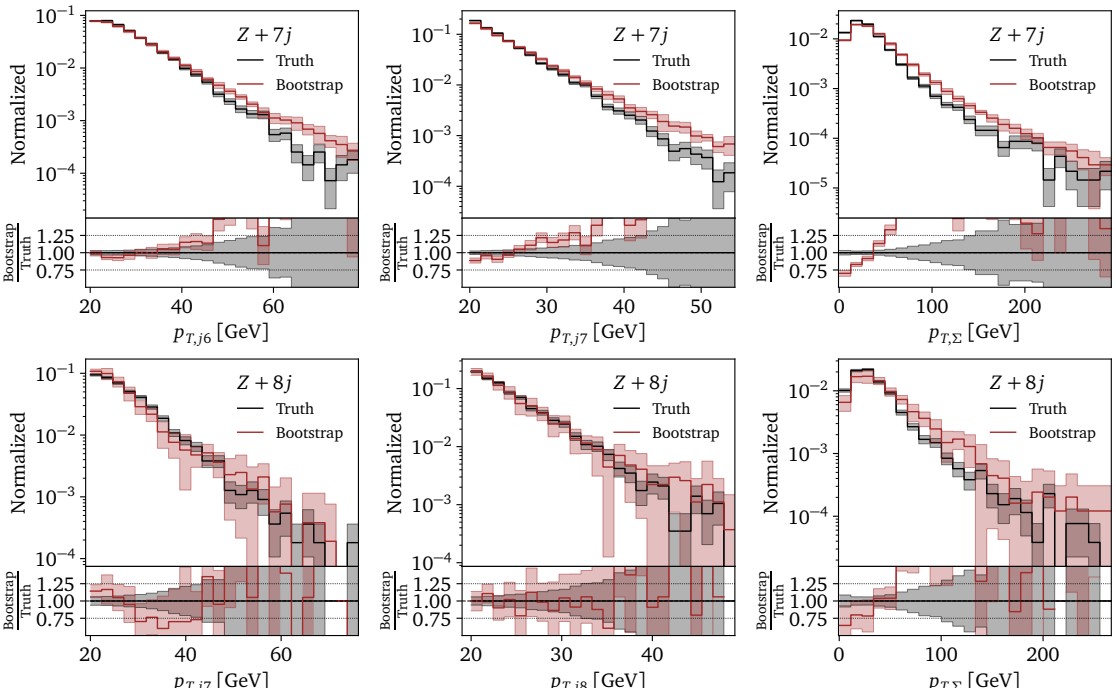

Figure 8: Selection of features in $Z + 7$ and 8-jet events for a generator trained on up to 6-jet events using the bootstrap technique.

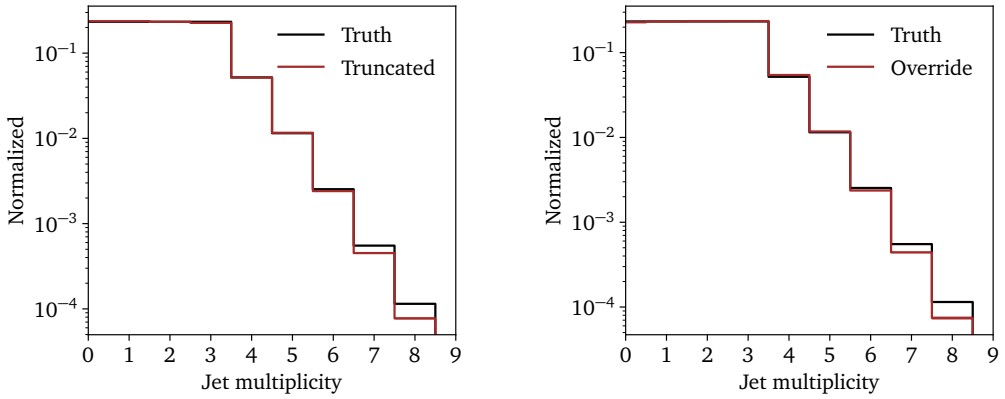

Figure 9: Jet multiplicity distributions learned using $\mathcal{L}_{\text{trunc}}$ (left) and $\mathcal{L}_{\text{override}}$ (right) trained on events with up to 6 jets.

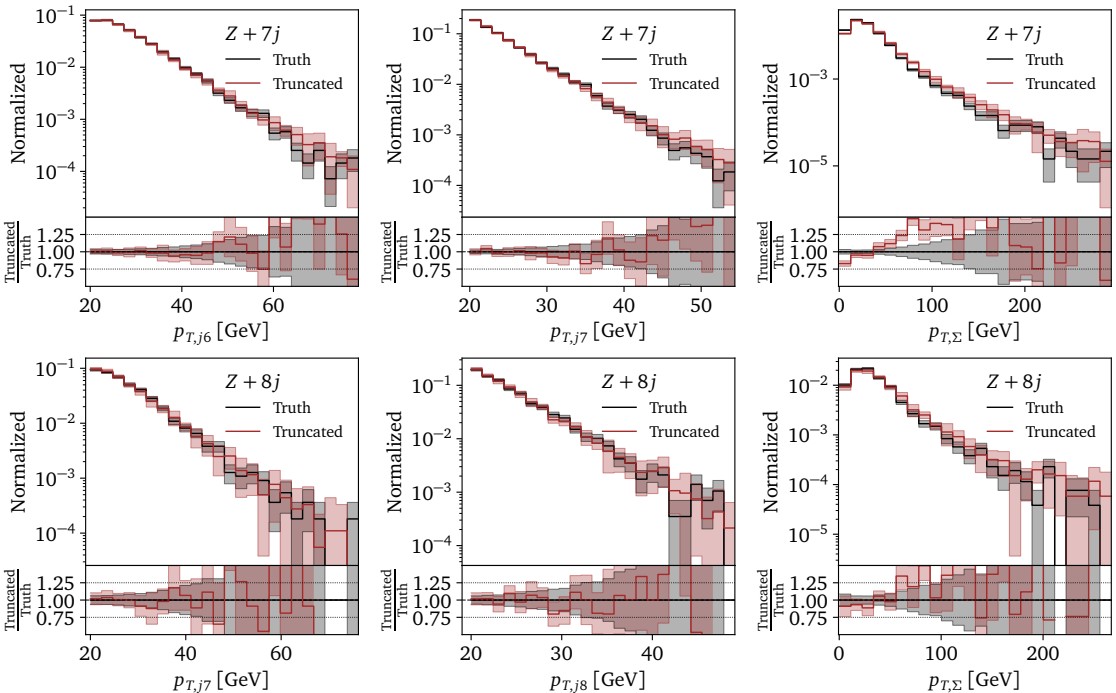

Figure 10: Selection of features $Z + 7$ and 8-jet events, trained with the truncated loss.

simple solution is to omit the final Bernoulli contribution from the loss and truncate the loss as described in Eq.(22),

$$\mathcal{L}_{\text{trunc}} = \left\langle -\sum_{i=1}^{n} \log p_{\text{kin}}(x_i|\nu_{i-1}) - \sum_{i=0}^{n} (1 - \delta_{in_{\max}}) \log p_{\text{bin}}(1 - \delta_{in}; \rho_i) \right\rangle_{x \sim p_{\text{data}}}, \qquad (25)$$

It differs from the complete likelihood loss of Eq.(21) in the addition of the factor $1 - \delta_{in_{\max}}$ in front of the Bernoulli component. Now, the splitting prediction for maximum-length events, $\rho_{n_{\max}}$, is not explicitly trained. Rather, the weight sharing in the transformer allows correlations learned at lower multiplicity to be recycled. When sampling a network trained in this way, the splitting predictions beyond $n_{\max}$ are pure extrapolation.

Using the truncated loss, we again train a transformer on events with up to 6 jets and again sample up to 8 jets. The generated multiplicities are shown in Fig. 9 (left). Indeed, the network learns and extrapolates the staircase scaling. We show the extrapolated kinematic correlations in Fig. 10. The only deviation exceeding the BNN uncertainty is a slightly larger transverse momentum imbalance than expected in 7-jet events. This result demonstrates that the generative transformer described in Sec. 2 has learned the universal pattern of jet radiation.

## 3.4 Extrapolation with override

In the previous section we have shown how truncating the final Bernoulli term from the likelihood loss allows the network to generate high-quality 7-jet and 8-jet events. However, the extrapolation can be mis-calibrated. Because the transformer splitting predictions $\rho_i$ are trained with a binary cross entropy, the optimal solution in terms of the non-splitting probability is the posterior

$$1 - \rho_i \approx p(\text{stop at } i|x_{1:i}) = \frac{p(x_{1:i}|\text{stop at } i)}{\sum_{n \geq i} p(x_{1:i}|\text{stop at } n)} \qquad (26)$$

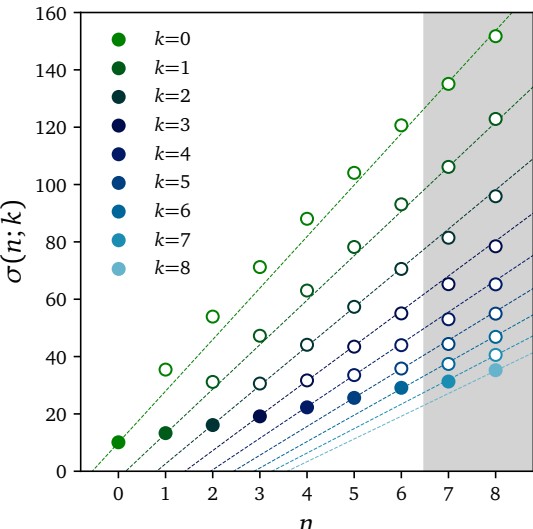

Figure 11: Standard deviations of $p_{x,\Sigma}$ for the muons and first $k$ jets in $Z + n$-jet events. Filled circles indicate complete events, with $k = n$, while empty circles are incomplete. Points in the gray region are not used in any fit, but show the agreement of the extrapolation.

assuming a uniform prior for simplicity. When training on a dataset with maximum multiplicity $n_{\max}$, the estimate is biased since the sum over $n$ is missing terms above $n_{\max}$. The same effect causes the transformer to stick to a constant splitting probability $\rho_{n_{\max}} = 0$.

In an alternative approach we show that transverse momentum conservation can be used as an extra handle on the posterior. A violation of transverse momentum conservation can be induced by removing particles beyond the hard process and first $k$ jets. The spread in center of momentum scales with $k$, so we can use transverse momentum conservation to statistically separate complete and incomplete events. Secondly, we note that the $p_{x,\Sigma}$ and $p_{y,\Sigma}$ distributions are roughly Gaussians with zero mean, and hence fully specified by their standard deviation.

In Fig. 11, we show the widths of the $p_{x,\Sigma}$ distributions as a function of the jet number, for complete events and for the hard process plus $k$ jets. The widths obey an approximately linear scaling when considering a fixed number of jets, for complete events or otherwise. We can perform a linear fit to estimate the standard deviations for higher-multiplicity events, giving analytic expressions for the likelihoods in Eq.(26). We arrive at

$$\sigma(n;k) = (n-k)m_k + \sigma(k;k)\,,$$

$$\text{with} \quad \sigma(k;k) = 3.14k + 9.97\,,$$
$$\text{and} \quad 1/m_k = 0.0088k + 0.056\,. \tag{27}$$

The widths of completed events, $\sigma(k;k)$, are fit from the bottom row of filled circles in Fig. 11 up to $n = 6$. The gradients $m_k$ of lines with constant $k$ are fit using events up to $n = 5$. The fits are shown as dotted lines in Fig. 11, and we see that they extrapolate well to all partial $k$ values in 7-jet and 8-jet events. Due to the rotation symmetry around the beam axis, the same values hold for $p_{y,\Sigma}$ and we can assume that the joint likelihoods are the product of 1D Gaussians.

$$p_{\text{fit}}(x_{1:k}|\text{stop at } n) = \mathcal{N}\left(p_{x,\Sigma}(x_{1:k})\,\big|\,0, \sigma(n;k)\right)\mathcal{N}\left(p_{y,\Sigma}(x_{1:k})\,\big|\,0, \sigma(n;k)\right) \tag{28}$$

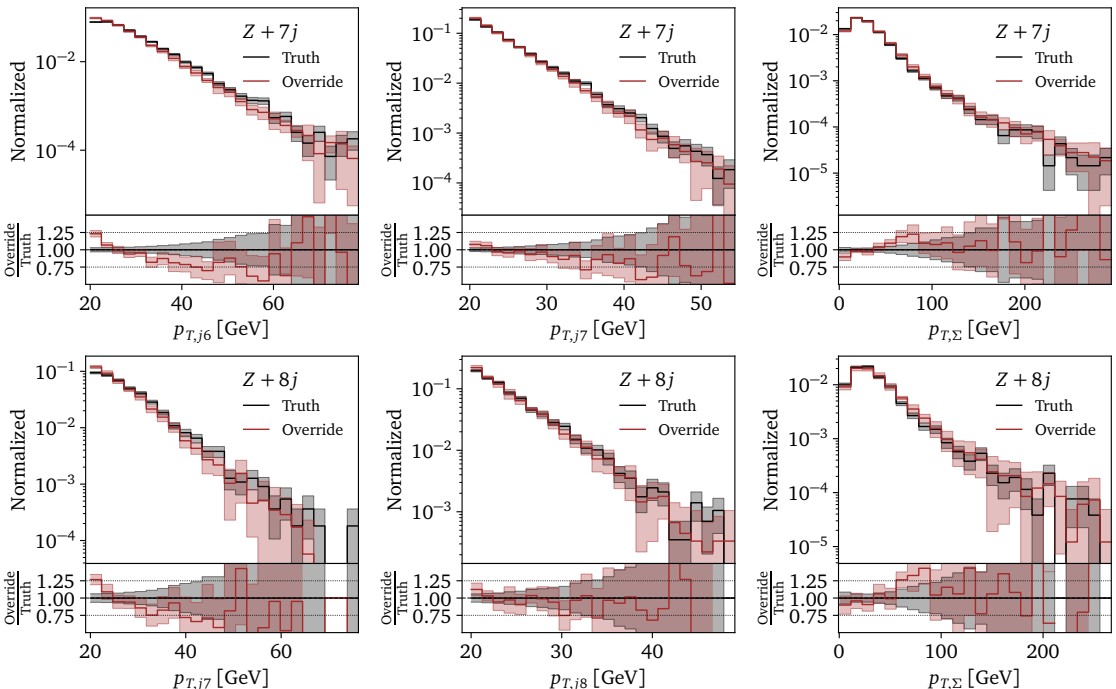

Figure 12: Selection of features $Z + 7$ and 8-jet events, trained with $\mathcal{L}_{\text{override}}$ on up to 6 jets.

Using Eq.(28) we can calculate target posteriors for an arbitrary maximum number of jets. This allows us to modify the likelihood loss by generalizing the Bernoulli splitting variable $\delta_{in}$ to a continuous variable $y_i \in [0,1]$ and override the troublesome $p_{\text{split}} = 0$ label for particle $n_{\max}$ by this estimate weighted by a hyperparameter $\lambda$,

$$\mathcal{L}_{\text{override}} = \left\langle -\sum_{i=1}^{n} \log p_{\text{kin}}(x_i | v_{i-1}) - \sum_{i=0}^{n} \lambda_i \log p_{\text{bin}}(y_i; \rho_i) \right\rangle_{x \sim p_{\text{data}}} \tag{29}$$

$$\text{with} \quad y_i(x_{1:n}) = \begin{cases} (1 - \delta_{in}) & i < n_{\max} \\ 1 - p_{\text{fit}}(\text{stop at } i | x_{1:i}) & i = n_{\max} \end{cases}, \tag{30}$$

$$\text{and} \quad \lambda_i = 1 - (1 - \lambda)\delta_{in_{\max}}. \tag{31}$$

The posterior $p_{\text{fit}}(\text{stop at } i | x_{1:i})$ is calculated using Eqs.(26) and (28) up to 8 jets. In practice, we find the best performance by including a staircase scaling prior when calculating the posterior. To match the dataset, we take $R_{(n+1)/n} = 0.225$ and cap the probabilities for $n < 3$. Note that the hyperparameter $\lambda$ multiplies only the $p_{\text{bin}}$ contribution for particle $n_{\max}$.

We train a network using the override loss with $\lambda = 0.2$ and sample events in the same manner as before ~~and show its event multiplicity distribution~~. In Fig. 9 (right), we show the event multiplicity distribution. ~~and network samples in Fig.~~ Similarly to the truncated loss, this override approach significantly increas~~ing~~es the fraction of higher-multiplicity compared to the naive extrapolation. Looking at the kinematics of the network samples, shown in Fig. 12, the $p_T$ distributions now display an excess toward low values, but the global momentum correlation $p_{T,\Sigma}$ is reproduced to greater accuracy ~~than previously~~ compared to the case with the truncated loss. This is to be expected, since the override loss is specifically designed to match the global momentum distribution. Once again, this demonstrates that autoregressive transformers can learn the universal nature of jet radiation.

**Code availability**

The code is available as part of the public Heidelberg hep-ml code and tutorial library. The dataset is available upon request.

# 4   Outlook

The universality of splitting kernels and jet ratios in QCD provides the perfect physics question to see if appropriate generative networks can extrapolate. As an example, we study $Z$+jets events and the established staircase scaling of the jet number. The same procedure can be applied to generate the full set of jet constituents.

We employ an autoregressive transformer to learn a factorized likelihood for events across varying jet multiplicity. This autoregressive transformer sequentially predicts the kinematics of an additional jet along with the probability to radiate it. When trained the standard way, the transformer learns the kinematics of up to the 6 jets included in the training data with high fidelity. It also produces a small number of higher-multiplicity events with reasonable kinematics.

The first path towards extrapolation is to modify the training data with bootstrapping. This approach is straight-forward to adapt the multiplicity distribution, but some kinematic distributions are not learned very precisely.

Another way to extrapolate is to truncate the loss function and remove the explicit learning of the hard Sudakov factor for the highest multiplicity. Alternatively, we can override the hard Sudakov factor using physics information. We find that ~~all~~ both of these approaches are equally capable of generating high-quality events. These results establish that autoregressive transformers can learn the universal nature of jet radiation.

We emphasize that our study only shows that generative networks can extrapolate, given the right QCD properties of the training data. We expect the performance of all extrapolating networks to improve with technical advances. One idea for such an improvement might be the synchronous training of a transformer generator with a classifier, as described in Appendix A.

# Acknowledgements

We thank Nathanael Ediger and Maeve Madigan for their contributions to earlier stages of this project, and Armand Rousselot and Sander Hummerich for valuable discussions on the DiscFormer. AB and JMV are funded by the BMBF Junior Group *Generative Precision Networks for Particle Physics* (DLR 01IS22079). JS is funded by the Carl-Zeiss-Stiftung through the project *Model-Based AI: Physical Models and Deep Learning for Imaging and Cancer Treatment*. The Heidelberg group is supported by the Deutsche Forschungsgemeinschaft (DFG, German Research Foundation) under grant 396021762 – TRR 257 *Particle Physics Phenomenology after the Higgs Discovery*. This work was also supported by the DFG under Germany's Excellence Strategy EXC 2181/1 - 390900948 *The Heidelberg STRUCTURES Excellence Cluster*. Moreover, we would like to thank the Baden-Württemberg Stiftung for financing through the program *Internationale Spitzenforschung*, project *Uncertainties – Teaching AI its Limits* (BWST_ISF2020-010). The authors acknowledge support by the state of Baden-Württemberg through bwHPC and the German Research Foundation (DFG) through grant no INST 39/963-1 FUGG (bwForCluster NEMO).

# A  Improving likelihood training with dynamic reweighting

The traditional way to achieve ultimate precision in distributions obtained via generative networks has been via reweighting. However, this comes at the cost of having weighted events. These weights typically span orders of magnitude which, in turn, translate into small efficiencies $\epsilon = \langle w \rangle / w_{\max}$ when performing accept-reject unweighting [110]. These inefficiencies imply a reduction in statistical power, and thus reweighting might render the original motivation of speeding up event generation invalid, since now one ~~has~~ might need to generate many more weighted events to obtain a similar number of unweighted events. One way to reduce event weights is to incorporate the discriminator information into the generator training. In the following sections we introduce and showcase the DiscFormer algorithm, inspired by the DiscFlow developed in Ref. [54] for normalizing flows.

## A.1  Dynamic discriminator reweighting

The main idea behind the dynamic discriminator reweighting is to extend the usual log-likelihood loss with an event-dependent weighting factor $w(x)$

$$\mathcal{L} = \langle -w^{\alpha}(x) \log p_{\theta}(x) \rangle_{x \sim p_{\text{data}}}. \tag{32}$$

where $\alpha > 0$ is a tunable hyperparameter and $p_{\theta}(x)$ is the learned likelihood that depends on learnable network parameters $\theta$. The weighting factor $w(x)$ approximates the likelihood ratio $p_{\text{data}}(x)/p_{\theta}(x)$ and is obtained from the score $D(x)$ of a neural discriminator. In practice, the weighting factor $w(x)$ in Eq.(32) amplifies the loss in regions where the generator is not precise enough. Intuitively, this can be thought of as modifying the training data distribution $p_{\text{data}}$ to increase the difference between $p_{\text{data}}(x)$ and $p_{\theta}(x)$. This discriminator transformation, or DiscFormation, is visualized in Fig. 13.

The neural discriminator $D(x)$ is trained to distinguish true samples, drawn from $p_{\text{data}}(x)$, from generated samples, drawn from $p_{\theta}(x)$

$$
\begin{aligned}
\mathcal{L} &= \langle -\log D(x) \rangle_{x \sim p_{\text{data}}} + \langle -\log(1 - D(x)) \rangle_{x \sim p_{\theta}} \\
&= -\int dx \, p_{\text{data}}(x) \log D(x) - \int dx \, p_{\theta}(x) \log(1 - D(x)) \, .
\end{aligned}
\tag{33}
$$

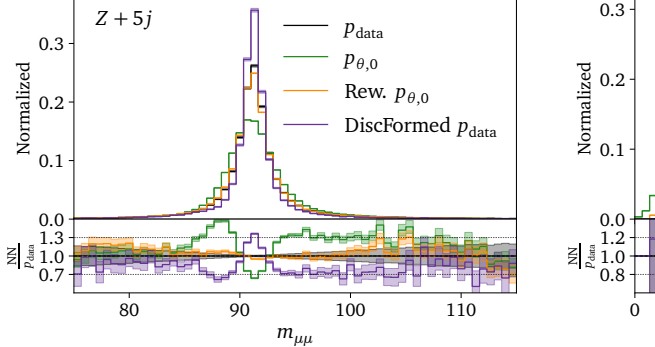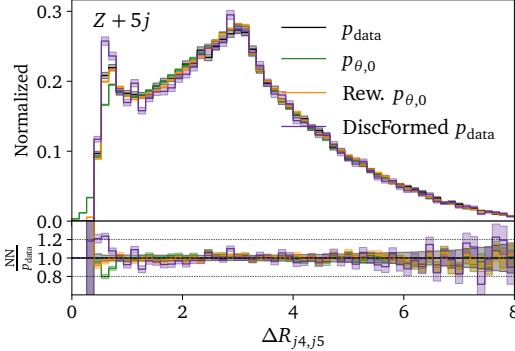

Figure 13: Distributions of the $Z$ boson mass (left) and $\Delta R_{j4,j5}$ (right) from the initial generator (green), the corresponding reweighted distribution $w_0 \cdot p_{\theta}(x)$ (orange) and the DiscFormed distribution $w_0 \cdot p_{\text{data}}(x)$ (purple).

We use variational calculus to find the minimum of this objective, yielding [54, 111]

$$0 = \frac{\delta \mathcal{L}}{\delta D(x)} = -\frac{p_{\text{data}}(x)}{D(x)} + \frac{p_\theta(x)}{1 - D(x)} \qquad \text{and} \qquad w(x) = \frac{D(x)}{1 - D(x)} = \frac{p_{\text{data}}(x)}{p_\theta(x)} \,. \tag{34}$$

The assumption that $w(x)$ correctly approximates the likelihood ratio $p_{\text{data}}(x)/p_\theta(x)$ can be validated by checking that the reweighted distributions correctly close onto a test set. We perform this test in Sec. A.4, and plot the resulting reweighted distribution as the orange histogram in Fig. 13.

With this assumption, we can prove that the extended generator loss (32) has the unique minimum $p_\theta(x) = p_{\text{data}}(x)$. To this end, we insert the perfect discriminator criterion (34) into the generator loss (32) and add a Lagrange multiplier term to enforce the normalization of the learned likelihood $p_\theta$, leading to the objective

$$\mathcal{L} = -\int dx \, p_{\text{data}}(x) \left( \frac{D(x)}{1 - D(x)} \right)^\alpha \log p_\theta(x) + \lambda \left( \int dx \, p_\theta(x) - 1 \right). \tag{35}$$

In real-world implementations the learned likelihood $p_\theta$ is constructed to satisfy the normalization constraint and the explicit Lagrange multiplier is not needed. For instance, we will construct $p_\theta$ as a product of Gaussian mixture models which are normalized by construction. We now use variational calculus to find the minimum of this objective, yielding

$$0 = \frac{\delta \mathcal{L}}{\delta p_\theta(x)} = -\left( \frac{D(x)}{1 - D(x)} \right)^\alpha \frac{p_{\text{data}}(x)}{p_\theta(x)} + \lambda = -\left( \frac{p_{\text{data}}(x)}{p_\theta(x)} \right)^{\alpha+1} + \lambda \,. \tag{36}$$

In the second equality we have again used the assumption of a perfectly trained discriminator, immediately yielding the unique solution $p_\theta(x) = p_{\text{data}}(x)$, and $\lambda = 1$. We shall now discuss the behavior of this solution for several values of $\alpha$:

- The usual log-likelihood loss $\alpha = 0$ emerges as a smooth decoupling limit where the discriminator weight does not contribute to the loss.

- In the limit $\alpha \to -1$, Eq. (36) becomes $-1 + \lambda = 0$ and the optimization problem has no longer a unique solution. This is easy to understand, as the loss becomes

$$\mathcal{L} = \langle -\log p_\theta(x) \rangle_{x \sim p_\theta} \,, \tag{37}$$

  where no information on $p_{\text{data}}(x)$ is included.

- For $\alpha + 1 < 0$, the second derivative of the loss becomes negative. This means we can no longer train $p_\theta(x)$ to approximate $p_{\text{data}}(x)$, since $p_\theta(x) = p_{\text{data}}(x)$ is now a maximum of the loss:

$$\frac{\delta^2 \mathcal{L}}{\delta p_\theta^2(x)} = (\alpha + 1) \frac{1}{p_{\text{data}}(x)} \left( \frac{p_{\text{data}}(x)}{p_\theta(x)} \right)^{\alpha+2} < 0 \quad \forall x. \tag{38}$$

In our experiments, we always set $\alpha = 1.0$ for the better training stability this choice offers, after checking that qualitatively the results behave in similar fashion for $0 < \alpha \lesssim 5$. In the subsections below, we describe in more detail how the DiscFormer training works, and demonstrate the capabilities of this approach in learning the full phase-space density of the $Z + 5$ jets dataset.

## A.2  Iterative DiscFormer algorithm

To train a generator with the DiscFormer approach described above, we require a discriminator $w(x) = p_{\text{data}}(x)/p_\theta(x)$ that tracks the state of the generator $p_\theta$ throughout the generator

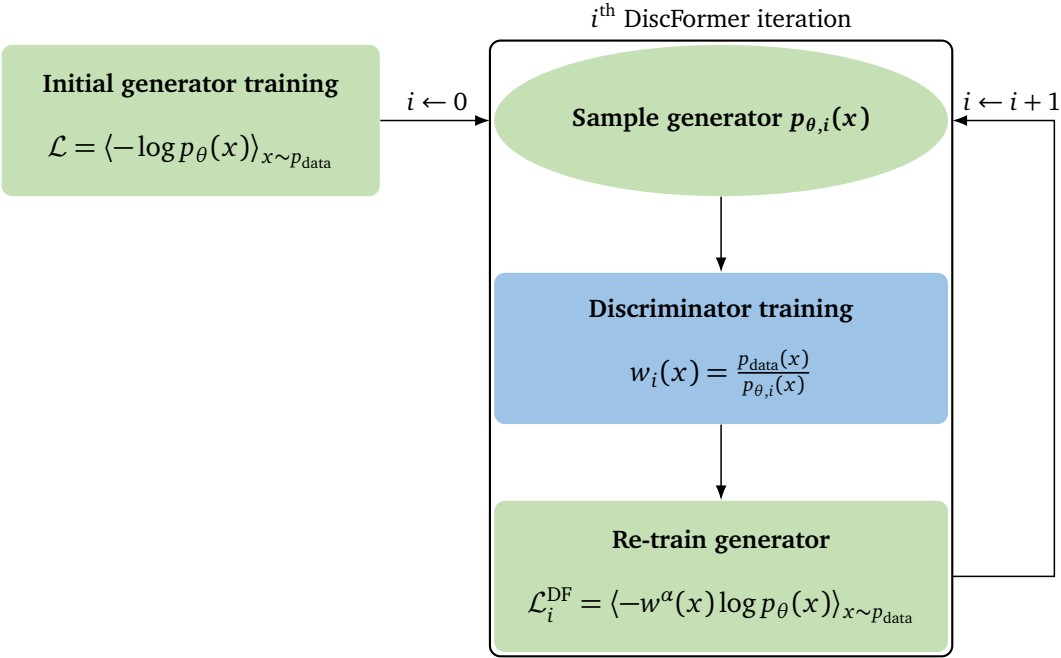

Figure 14: The iterative implementation of the DiscFormer algorithm. The re-training of the generator is done with the loss detailed in Eq.(32).

training. Clearly, training such a discriminator for each generator update step is not a scalable approach. Ref. [54] proposed a GAN-like approach where the generator and discriminator are trained jointly. Similar to GANs, this approach requires careful hyperparameter tuning to train the two networks jointly.

We propose an iterative procedure using discriminator checkpoints to avoid a joint training. To this end, we extract a checkpoint generator $p_{\theta,0}(x)$ and use it to train a discriminator $D_0(x)$ to obtain the weights $w_0(x)$. Using the saved likelihoods of the checkpoint generator $p_{\theta,0}(x)$ and the likelihoods of the current generator state $p_\theta(x)$, we can reweight the checkpoint weights $w_0(x)$ to obtain the full weights $w(x)$

$$w_0(x) = \frac{p_{\text{data}}(x)}{p_{\theta,0}(x)} = \frac{D_0(x)}{1 - D_0(x)}$$

$$w(x) = \frac{p_{\text{data}}(x)}{p_\theta(x)} = \frac{p_{\text{data}}(x)}{p_{\theta,0}(x)} \frac{p_{\theta,0}(x)}{p_\theta(x)} = w_0(x) \frac{p_{\theta,0}(x)}{p_\theta(x)} . \tag{39}$$

We emphasize that this checkpoint reweighting is exact, thanks to the fact that we can extract likelihoods from the generative model. The only approximation is in the assumption that the checkpoint discriminator $D_0(x)$ learns the likelihood ratio perfectly. Whenever the generator $p_\theta(x)$ approximates the truth distribution $p_{\text{data}}(x)$ significantly better than the checkpoint generator $p_{\theta,0}(x)$, it is important that we update the checkpoint generator to $p_{\theta,1}(x)$ to avoid numerical issues from cancellations between $w_0(x)$ and $p_{\theta,0}(x)/p_\theta(x)$. This leads us to the loss function in the $i^{\text{th}}$ DiscFormer iteration

$$\mathcal{L}_i^{\text{DF}} = \langle -w^\alpha(x) \log p_\theta(x) \rangle_{x \sim p_{\text{data}}} = \left\langle -\left( \frac{D_i(x)}{1 - D_i(x)} \frac{p_{\theta,i}(x)}{p_\theta(x)} \right)^\alpha \log p_\theta(x) \right\rangle_{x \sim p_{\text{data}}} . \tag{40}$$

The implementation of the DiscFormer algorithm is depicted in Fig. 14. We will describe now how the first iteration in the algorithm takes place. The process starts with an initial

training of the generator, until its convergence is reached at state $p_{\theta,0}(x)$. From this point, we start the DiscFormer iteration, represented as a black rectangle in Fig. 14. We first draw samples from the generator, and train the discriminator to learn the likelihood ratio of $p_{\text{data}}(x)$ to $p_{\theta,0}(x)$. To conclude the DiscFormer iteration, we re-train the generator using the DiscFormer loss, warm-starting it from $p_{\theta,0}(x)$. This procedure can be repeated until convergence.

### A.3   Autoregressive transformer

The DiscFormer formalism developed above can be applied to any generative network that allows us to extract likelihoods $p_\theta(x)$. Particularly attractive approaches are normalizing flows and parametric models like the generative transformer used in the main body of this work due to their ability of fast likelihood evaluation. We use an autoregressive transformer [28], which is designed to interpret the phase space vector $x = (x_1, \ldots, x_n)$ as a sequence of elements, and factorizes the $n$-dimensional probability into $n$ probabilities, successively conditioned

$$p_\theta(x) = \prod_{i=1}^{n} p_\theta(x_i | x_1, ..., x_{i-1}), \tag{41}$$

The sequence elements are the kinematical quantities $(p_T, \phi, \eta, m)$, preprocessed as described in Sec. 2.2. This autoregressive approach is mainly beneficial to our case because we can use our physics knowledge to group challenging phase space directions early in the sequence $x_1, ..., x_n$. In contrast to the autoregressive transformer used in the main body of this work, this approach does not include splittings.

The network learns the factorizing conditional probabilities over phase space using a Gaussian mixture representation $\mathcal{G}$:

$$p_\theta(x_i | \omega^{(i-1)}) = \sum_{G_j \in \mathcal{G}} w_j^{(i-1)} \mathcal{N}(x_i; \mu_j^{(i-1)}, \sigma_j^{(i-1)}), \tag{42}$$

where $\{w_j, \mu_j, \sigma_j\}$ are the components of the $j$-th gaussian. In terms of architecture, our generator contains 80k parameters, consisting of 2 transformer decoder blocks, 4 self-attention heads and 50 Gaussian mixture elements. For more details, we refer the reader to Tab. 4. The network is trained on approximately 80k and tested on 40k events from the $Z + 5$ jets dataset. We choose this working point to demonstrate the capabilities the DiscFormer approach can have when training data is scarce, compared to standard log-likelihood loss training. The challenges that the autoregressive generator faces for the $Z + 5$ jets dataset are: achieving percent level precision in the $Z$ mass peak and resolving the several hard $\Delta R$ boundaries between jets. For the latter, 5 jets originate a total of $5(5-1)/2$ distinct $\Delta R$ features and the generator has to learn the subtle differences between them. The feature ordering in the sequence $x_1, ..., x_n$ plays here a significant role in obtaining good performance for these features. We find that the best precision is achieved with

$$\{x_i\} = \{\phi, p_T, \eta, m\}_{j_5, j_4, j_3, j_2, j_1} \cup \{\phi, p_T, \eta\}_{\mu_2, \mu_1}. \tag{43}$$

We check that the $Z$ mass peak precision, on the other hand, is affected less by changes in the feature ordering.

### A.4   Results

**Discriminator reweighting**

In the derivation of the DiscFormer loss, we have made the assumption that $D(x)$ is a discriminator that correctly approximates the likelihood ratio of $p_{\text{data}}(x)/p_\theta(x)$. We demonstrate in

| Run \ AUC | $p_{\theta,0}(x)$ | $p_{\theta,3}^{\mathrm{DF}}(x)$ | $p_{\theta,3}(x)$ |
|---|---|---|---|
| 1 | 0.737 | 0.712 | 0.743 |
| 2 | 0.754 | 0.711 | 0.766 |
| 3 | 0.695 | 0.639 | 0.682 |
| 4 | 0.701 | 0.675 | 0.686 |
| 5 | 0.713 | 0.633 | 0.704 |
| 6 | 0.696 | 0.651 | 0.710 |
| 7 | 0.699 | 0.664 | 0.671 |
| 8 | 0.730 | 0.657 | 0.680 |
| 9 | 0.676 | 0.629 | 0.707 |
| avg. $\pm$ std | $0.711 \pm 0.023$ | $0.663 \pm 0.029$ | $0.706 \pm 0.029$ |

Table 2: AUC values as quality metrics to study the DiscFormer performance, evaluated on 9 independent seeds. We show, from left to right, the initial generator $p_{\theta,0}$, the final generator $p_{\theta,3}^{\mathrm{DF}}(x)$ after 3 discformer iterations, and the generator $p_{\theta,3}(x)$ trained for the same number of epochs with a standard likelihood loss.

this subsection that the discriminator network is indeed capable of identifying failure modes of the generator $p_\theta(x)$. Our discriminator is a transformer with 4 transformer-decoder blocks and 4 self-attention heads, summing up to about 800k parameters. We use the kinematic quantities $(p_T, \phi, \eta, m)$ for the final-state particles and the virtual $Z$ boson as input tokens, preprocessed in the same way as described in Sec. 2.2. Additionally, we add tokens for the pairwise $\Delta R$ to inform the discriminator about this challenging correlation. Each time we train the discriminator, we do so with early stopping and a patience of 10 epochs. After each training, we check that the discriminator learns the correct likelihood ratio by reweighting the generated samples and checking that they close onto a test set. In Fig. 13 we show such closure test for the iteration 0 of one DiscFormer run. In this case, we find that the reweighted distribution $w_0(x) \cdot p_{\theta,0}(x)$ correctly matches the truth.

**DiscFormer**

Finally, we discuss the details of the DiscFormer experiment, where we modify the standard likelihood loss by incorporating discriminator information during training. The setup described below is designed such that we can compare the results from the DiscFormer approach directly to the standard likelihood loss training:

1. To start, we train the initial generator for 2500 epochs and extract the network with the best validation loss. We name this generator $p_{\theta,0}(x)$.
2. From the $p_{\theta,0}(x)$ checkpoint, we compare two options: DiscFormer iterations, and standard training.
   - DiscFormer trainings cover 3 iterations, each consisting of generator sampling, discriminator training and generator training with DiscFormer loss. The discriminator is trained from scratch at each iteration with early stopping and a patience of 10 epochs. The discriminator with best validation loss is also loaded for evaluation. The generator, on the other hand, is warm-started from the final state of the previous iteration and trained for 500 epochs. We call the final state of the generator at the end of the run $p_{\theta,3}^{\mathrm{DF}}(x)$.
   - Standard generator trainings are continued from the same $p_{\theta,0}(x)$ checkpoint for the same number of epochs as in the DiscFormer algorithm, i.e. $3 \times 500 = 1500$. We name this generator $p_{\theta,3}(x)$.

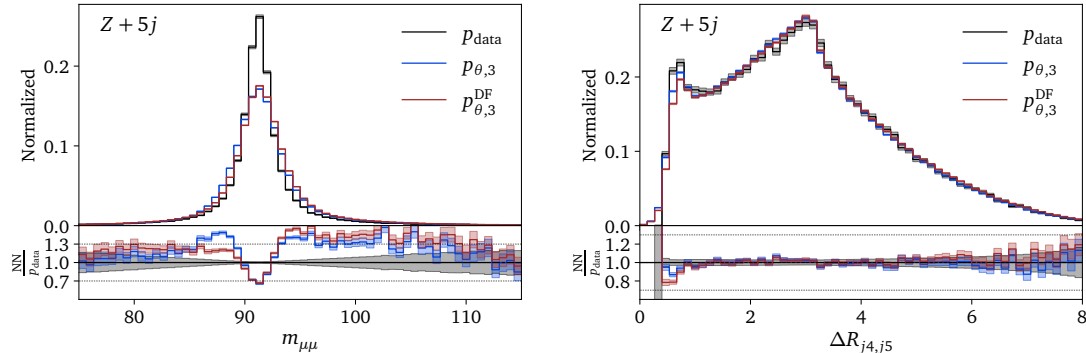

Figure 15: Distributions of the $Z$ boson mass (left) and $\Delta R_{j4,j5}$ (right) obtained from the standard generator training (blue) and from the generator after training for 3 DiscFormer iterations (red). Both networks have been trained for the same number of epochs.

The final states of the generator $p_{\theta,3}^{\text{DF}}(x)$ and $p_{\theta,3}(x)$ are thus directly comparable, in the sense that they have been trained for the same number of epochs, and differences in their performance can be, in principle, attributed to the different training modalities. We find that the performance of the DiscFormer approach saturates after 3 iterations.

To systematically check whether performance is gained through the DiscFormer approach, we perform 9 identical runs where we compare the DiscFormer training versus the standard training. For these runs, we train discriminators with identical architecture to distinguish true data from samples generated from the corresponding $p_\theta(x)$, and show the Area Under the Curve (AUC) of the Receiver-Operating Characteristic (ROC) curve obtained from evaluating these classifiers on a test set in Tab. 2. We find that the discriminators trained to distinguish $p_{\theta,3}^{\text{DF}}(x)$ from truth data have systematically lower AUC values than those trained to distinguish $p_{\theta,3}(x)$ from truth. In particular, we observe that standard likelihood training improves the quality of the samples only marginally according to the discriminators, i.e. $p_{\theta,0}(x)$ and $p_{\theta,3}(x)$ showing very similar AUC values, whereas we observe a systematic improvement for the DiscFormer training $p_{\theta,3}^{\text{DF}}(x)$.

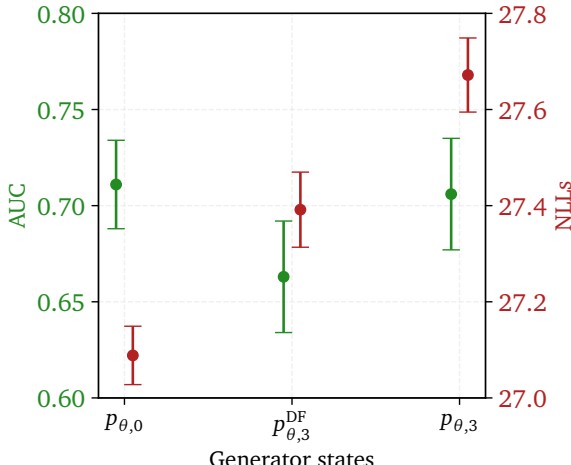

Figure 16: Mean and standard deviation of the AUC and negative log-likelihood (NLL) values evaluated on the same test set for the 9 runs from Tab. 2.

We can also check the performance of $p_{\theta,3}(x)$ and $p_{\theta,3}^{\text{DF}}(x)$ on two challenging phase space features, the $Z$ mass and $\Delta R_{j4,j5}$, shown in Fig. 15. We observe that the generator trained with the DiscFormer approach performs slightly better in some of the regions near the peak of the $m_Z$ distribution, whereas the generator trained with the standard likelihood loss has become better in the tails. On the other hand, the $\Delta R_{j4,j5}$ sharp boundary seems to be slightly better resolved by the standard generator than by the generator trained with DiscFormer loss. More generally, we find no systematic performance differences in those phase space distributions.

As a third metric, we evaluate the negative log-likelihood (NLL) of all 3 generator stages on the same test set, and compute the mean and the standard deviation for the same 9 runs shown in Tab. 2. These, along with the mean and standard deviation of the AUC values, are shown in Fig. 16. We observe that the best NLL is generally obtained for the initial generator $p_{\theta,0}(x)$. This makes sense, as this network was trained to minimize the NLL until the validation loss started to increase. In contrast, $p_{\theta,3}(x)$ was trained beyond that point, leading to larger NLLs for the test data. The DiscFormer $p_{\theta,3}^{\text{DF}}(x)$ was trained as well beyond the minimum but with the target to minimize the DiscFormer loss. Nonetheless, the NLL is significantly better for the generator trained with the DiscFormer approach than for the generator trained with standard likelihood loss.

## B Network hyperparameters

| Parameter | Value |
|---|---|
| Optimizer | Adam |
| Learning rate | $3 \cdot 10^{-4}$ |
| LR schedule | constant |
| Batch size | 512 |
| # Iterations | 200k |
| # Transformer Blocks | 3+3 |
| Latent space size $d$ | 128 |
| # Attention heads | 8 |
| # Mixture model elements | 42 |
| # Trainable parameters | 1.2M |

Table 3: Architecture and training hyperparameters. We use 3 blocks each for the particle-level transformer and the component-level transformer.

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

| Parameter | Generator | Discriminator |
|---|---|---|
| Optimizer | Adam | Adam |
| Learning rate | $1 \cdot 10^{-4}$ | $3 \cdot 10^{-4}$ |
| LR schedule (initial) | OneCycleLR | – |
| LR schedule (DiscFormer) | constant | ReduceLROnPlateau |
| Batch size | 512 | 512 |
| # Iterations (initial) | until early stopping | – |
| # Iterations (DiscFormer) | 5×75k | until early stopping |
| # Transformer Blocks | 2 | 4 |
| # Attention heads | 4 | 4 |
| # Mixture model elements | 50 | – |
| # Trainable parameters | 80k | 800k |

Table 4: Training hyperparameters and architecture for DiscFormer generator and discriminator.

[4] C. Gao, J. Isaacson, and C. Krause, *i-flow: High-dimensional Integration and Sampling with Normalizing Flows*, Mach. Learn. Sci. Tech. **1** (1, 2020) 045023, arXiv:2001.05486 [physics.comp-ph].

[5] C. Gao, S. Höche, J. Isaacson, C. Krause, and H. Schulz, *Event Generation with Normalizing Flows*, Phys. Rev. D **101** (2020) 7, 076002, arXiv:2001.10028 [hep-ph].

[6] K. Danziger, T. Janßen, S. Schumann, and F. Siegert, *Accelerating Monte Carlo event generation – rejection sampling using neural network event-weight estimates*, SciPost Phys. **12** (9, 2022) 164, arXiv:2109.11964 [hep-ph].

[7] T. Heimel, R. Winterhalder, A. Butter, J. Isaacson, C. Krause, F. Maltoni, O. Mattelaer, and T. Plehn, *MadNIS - Neural multi-channel importance sampling*, SciPost Phys. **15** (2023) 4, 141, arXiv:2212.06172 [hep-ph].

[8] E. Bothmann, T. Childers, W. Giele, F. Herren, S. Hoeche, J. Isaacsson, M. Knobbe, and R. Wang, *Efficient phase-space generation for hadron collider event simulation*, SciPost Phys. **15** (2023) 169, arXiv:2302.10449 [hep-ph].

[9] T. Heimel, N. Huetsch, F. Maltoni, O. Mattelaer, T. Plehn, and R. Winterhalder, *The MadNIS reloaded*, SciPost Phys. **17** (2024) 1, 023, arXiv:2311.01548 [hep-ph].

[10] N. Deutschmann and N. Götz, *Accelerating HEP simulations with Neural Importance Sampling*, JHEP **03** (2024) 083, arXiv:2401.09069 [hep-ph].

[11] T. Heimel, O. Mattelaer, T. Plehn, and R. Winterhalder, *Differentiable MadNIS-Lite*, arXiv:2408.01486 [hep-ph].

[12] F. Bishara and M. Montull, *(Machine) Learning Amplitudes for Faster Event Generation*, arXiv:1912.11055 [hep-ph].

[13] S. Badger and J. Bullock, *Using neural networks for efficient evaluation of high multiplicity scattering amplitudes*, JHEP **06** (2020) 114, arXiv:2002.07516 [hep-ph].

[14] J. Aylett-Bullock, S. Badger, and R. Moodie, *Optimising simulations for diphoton production at hadron colliders using amplitude neural networks*, JHEP **08** (6, 2021) 066, arXiv:2106.09474 [hep-ph].

[15] D. Maître and H. Truong, *A factorisation-aware Matrix element emulator*, JHEP **11** (7, 2021) 066, arXiv:2107.06625 [hep-ph].

[16] R. Winterhalder, V. Magerya, E. Villa, S. P. Jones, M. Kerner, A. Butter, G. Heinrich, and T. Plehn, *Targeting multi-loop integrals with neural networks*, SciPost Phys. **12** (2022) 4, 129, arXiv:2112.09145 [hep-ph].

[17] S. Badger, A. Butter, M. Luchmann, S. Pitz, and T. Plehn, *Loop amplitudes from precision networks*, SciPost Phys. Core **6** (2023) 034, arXiv:2206.14831 [hep-ph].

[18] D. Maître and H. Truong, *One-loop matrix element emulation with factorisation awareness*, arXiv:2302.04005 [hep-ph].

[19] J. Spinner, V. Bresó, P. de Haan, T. Plehn, J. Thaler, and J. Brehmer, *Lorentz-Equivariant Geometric Algebra Transformers for High-Energy Physics*, arXiv:2405.14806 [physics.data-an].

[20] D. Maître, V. S. Ngairangbam, and M. Spannowsky, *Optimal Equivariant Architectures from the Symmetries of Matrix-Element Likelihoods*, arXiv:2410.18553 [hep-ph].

[21] J. Brehmer, V. Bresó, P. de Haan, T. Plehn, H. Qu, J. Spinner, and J. Thaler, *A Lorentz-Equivariant Transformer for All of the LHC*, arXiv:2411.00446 [hep-ph].

[22] V. Bresó, G. Heinrich, V. Magerya, and A. Olsson, *Interpolating amplitudes*, arXiv:2412.09534 [hep-ph].

[23] J. M. Villadamigo, R. Frederix, T. Plehn, T. Vitos, and R. Winterhalder, *FASTColor – Full-color Amplitude Surrogate Toolkit for QCD*, arXiv:2509.07068 [hep-ph].

[24] B. Hashemi, N. Amin, K. Datta, D. Olivito, and M. Pierini, *LHC analysis-specific datasets with Generative Adversarial Networks*, arXiv:1901.05282 [hep-ex].

[25] R. Di Sipio, M. Faucci Giannelli, S. Ketabchi Haghighat, and S. Palazzo, *DijetGAN: A Generative-Adversarial Network Approach for the Simulation of QCD Dijet Events at the LHC*, JHEP **08** (2019) 110, arXiv:1903.02433 [hep-ex].

[26] A. Butter, T. Plehn, and R. Winterhalder, *How to GAN LHC Events*, SciPost Phys. **7** (2019) 6, 075, arXiv:1907.03764 [hep-ph].

[27] Y. Alanazi, N. Sato, T. Liu, W. Melnitchouk, M. P. Kuchera, E. Pritchard, M. Robertson, R. Strauss, L. Velasco, and Y. Li, *Simulation of electron-proton scattering events by a Feature-Augmented and Transformed Generative Adversarial Network (FAT-GAN)*, arXiv:2001.11103 [hep-ph].

[28] A. Butter, N. Huetsch, S. Palacios Schweitzer, T. Plehn, P. Sorrenson, and J. Spinner, *Jet Diffusion versus JetGPT – Modern Networks for the LHC*, arXiv:2305.10475 [hep-ph].

[29] M. Paganini, L. de Oliveira, and B. Nachman, *Accelerating Science with Generative Adversarial Networks: An Application to 3D Particle Showers in Multilayer Calorimeters*, Phys. Rev. Lett. **120** (2018) 4, 042003, arXiv:1705.02355 [hep-ex].

[30] M. Paganini, L. de Oliveira, and B. Nachman, *CaloGAN : Simulating 3D high energy particle showers in multilayer electromagnetic calorimeters with generative adversarial networks*, Phys. Rev. D **97** (2018) 1, 014021, arXiv:1712.10321 [hep-ex].

[31] M. Erdmann, J. Glombitza, and T. Quast, *Precise simulation of electromagnetic calorimeter showers using a Wasserstein Generative Adversarial Network*, Comput. Softw. Big Sci. **3** (2019) 1, 4, arXiv:1807.01954 [physics.ins-det].

[32] D. Belayneh *et al.*, *Calorimetry with Deep Learning: Particle Simulation and Reconstruction for Collider Physics*, Eur. Phys. J. C **80** (12, 2020) 688, arXiv:1912.06794 [physics.ins-det].

[33] E. Buhmann, S. Diefenbacher, E. Eren, F. Gaede, G. Kasieczka, A. Korol, and K. Krüger, *Getting High: High Fidelity Simulation of High Granularity Calorimeters with High Speed*, Comput. Softw. Big Sci. **5** (2021) 1, 13, arXiv:2005.05334 [physics.ins-det].

[34] C. Krause and D. Shih, *CaloFlow: Fast and Accurate Generation of Calorimeter Showers with Normalizing Flows*, arXiv:2106.05285 [physics.ins-det].

[35] ATLAS Collaboration, *AtlFast3: the next generation of fast simulation in ATLAS*, Comput. Softw. Big Sci. **6** (2022) 7, arXiv:2109.02551 [hep-ex].

[36] C. Krause and D. Shih, *CaloFlow II: Even Faster and Still Accurate Generation of Calorimeter Showers with Normalizing Flows*, arXiv:2110.11377 [physics.ins-det].

[37] E. Buhmann, S. Diefenbacher, D. Hundhausen, G. Kasieczka, W. Korcari, E. Eren, F. Gaede, K. Krüger, P. McKeown, and L. Rustige, *Hadrons, better, faster, stronger*, Mach. Learn. Sci. Tech. **3** (2022) 2, 025014, arXiv:2112.09709 [physics.ins-det].

[38] C. Chen, O. Cerri, T. Q. Nguyen, J. R. Vlimant, and M. Pierini, *Analysis-Specific Fast Simulation at the LHC with Deep Learning*, Comput. Softw. Big Sci. **5** (2021) 1, 15.

[39] V. Mikuni and B. Nachman, *Score-based generative models for calorimeter shower simulation*, Phys. Rev. D **106** (2022) 9, 092009, arXiv:2206.11898 [hep-ph].

[40] J. C. Cresswell, B. L. Ross, G. Loaiza-Ganem, H. Reyes-Gonzalez, M. Letizia, and A. L. Caterini, *CaloMan: Fast generation of calorimeter showers with density estimation on learned manifolds*, in *36th Conference on Neural Information Processing Systems*. 11, 2022. arXiv:2211.15380 [hep-ph].

[41] S. Diefenbacher, E. Eren, F. Gaede, G. Kasieczka, C. Krause, I. Shekhzadeh, and D. Shih, *L2LFlows: Generating High-Fidelity 3D Calorimeter Images*, arXiv:2302.11594 [physics.ins-det].

[42] B. Hashemi, N. Hartmann, S. Sharifzadeh, J. Kahn, and T. Kuhr, *Ultra-high-granularity detector simulation with intra-event aware generative adversarial network and self-supervised relational reasoning*, Nature Commun. **15** (2024) 1, 5825, arXiv:2303.08046 [physics.ins-det].

[43] A. Xu, S. Han, X. Ju, and H. Wang, *Generative Machine Learning for Detector Response Modeling with a Conditional Normalizing Flow*, arXiv:2303.10148 [hep-ex].

[44] E. Buhmann, S. Diefenbacher, E. Eren, F. Gaede, G. Kasieczka, A. Korol, W. Korcari, K. Krüger, and P. McKeown, *CaloClouds: Fast Geometry-Independent Highly-Granular Calorimeter Simulation*, arXiv:2305.04847 [physics.ins-det].

[45] M. R. Buckley, C. Krause, I. Pang, and D. Shih, *Inductive simulation of calorimeter showers with normalizing flows*, Phys. Rev. D **109** (2024) 3, 033006, arXiv:2305.11934 [physics.ins-det].

[46] S. Diefenbacher, V. Mikuni, and B. Nachman, *Refining Fast Calorimeter Simulations with a Schrödinger Bridge*, arXiv:2308.12339 [physics.ins-det].

[47] F. Ernst, L. Favaro, C. Krause, T. Plehn, and D. Shih, *Normalizing Flows for High-Dimensional Detector Simulations*, arXiv:2312.09290 [hep-ph].

[48] L. Favaro, A. Ore, S. P. Schweitzer, and T. Plehn, *CaloDREAM – Detector Response Emulation via Attentive flow Matching*, arXiv:2405.09629 [hep-ph].

[49] T. Buss, F. Gaede, G. Kasieczka, C. Krause, and D. Shih, *Convolutional L2LFlows: Generating Accurate Showers in Highly Granular Calorimeters Using Convolutional Normalizing Flows*, arXiv:2405.20407 [physics.ins-det].

[50] G. Quétant, J. A. Raine, M. Leigh, D. Sengupta, and T. Golling, *PIPPIN: Generating variable length full events from partons*, arXiv:2406.13074 [hep-ph].

[51] O. Amram *et al.*, *CaloChallenge 2022: A Community Challenge for Fast Calorimeter Simulation*, arXiv:2410.21611 [cs.LG].

[52] A. Butter, S. Diefenbacher, G. Kasieczka, B. Nachman, and T. Plehn, *GANplifying event samples*, SciPost Phys. **10** (2021) 6, 139, arXiv:2008.06545 [hep-ph].

[53] S. Bieringer, A. Butter, S. Diefenbacher, E. Eren, F. Gaede, D. Hundhausen, G. Kasieczka, B. Nachman, T. Plehn, and M. Trabs, *Calomplification — the power of generative calorimeter models*, JINST **17** (2022) 09, P09028, arXiv:2202.07352 [hep-ph].

[54] A. Butter, T. Heimel, S. Hummerich, T. Krebs, T. Plehn, A. Rousselot, and S. Vent, *Generative networks for precision enthusiasts*, SciPost Phys. **14** (2023) 4, 078, arXiv:2110.13632 [hep-ph].

[55] R. Winterhalder, M. Bellagente, and B. Nachman, *Latent Space Refinement for Deep Generative Models*, arXiv:2106.00792 [stat.ML].

[56] B. Nachman and R. Winterhalder, *ELSA – Enhanced latent spaces for improved collider simulations*, arXiv:2305.07696 [hep-ph].

[57] M. Leigh, D. Sengupta, J. A. Raine, G. Quétant, and T. Golling, *PC-Droid: Faster diffusion and improved quality for particle cloud generation*, arXiv:2307.06836 [hep-ex].

[58] R. Das, L. Favaro, T. Heimel, C. Krause, T. Plehn, and D. Shih, *How to understand limitations of generative networks*, SciPost Phys. **16** (2024) 1, 031, arXiv:2305.16774 [hep-ph].

[59] M. Bellagente, A. Butter, G. Kasieczka, T. Plehn, and R. Winterhalder, *How to GAN away Detector Effects*, SciPost Phys. **8** (2020) 4, 070, arXiv:1912.00477 [hep-ph].

[60] M. Bellagente, A. Butter, G. Kasieczka, T. Plehn, A. Rousselot, R. Winterhalder, L. Ardizzone, and U. Köthe, *Invertible Networks or Partons to Detector and Back Again*, SciPost Phys. **9** (2020) 074, arXiv:2006.06685 [hep-ph].

[61] M. Backes, A. Butter, M. Dunford, and B. Malaescu, *An unfolding method based on conditional Invertible Neural Networks (cINN) using iterative training*, arXiv:2212.08674 [hep-ph].

[62] M. Leigh, J. A. Raine, K. Zoch, and T. Golling, *ν-flows: Conditional neutrino regression*, SciPost Phys. **14** (2023) 6, 159, arXiv:2207.00664 [hep-ph].

[63] J. A. Raine, M. Leigh, K. Zoch, and T. Golling, *$v^2$-Flows: Fast and improved neutrino reconstruction in multi-neutrino final states with conditional normalizing flows*, arXiv:2307.02405 [hep-ph].

[64] A. Shmakov, K. Greif, M. Fenton, A. Ghosh, P. Baldi, and D. Whiteson, *End-To-End Latent Variational Diffusion Models for Inverse Problems in High Energy Physics*, arXiv:2305.10399 [hep-ex].

[65] S. Diefenbacher, G.-H. Liu, V. Mikuni, B. Nachman, and W. Nie, *Improving generative model-based unfolding with Schrödinger bridges*, Phys. Rev. D **109** (2024) 7, 076011, arXiv:2308.12351 [hep-ph].

[66] N. Huetsch, J. Mariño Villadamigo, A. Shmakov, S. Diefenbacher, V. Mikuni, T. Heimel, M. J. Fenton, K. T. Greif, B. Nachman, D. Whiteson, A. Butter, and T. Plehn, *The Landscape of Unfolding with Machine Learning*, arXiv:2404.18807 [hep-ph].

[67] S. Bieringer, A. Butter, T. Heimel, S. Höche, U. Köthe, T. Plehn, and S. T. Radev, *Measuring QCD Splittings with Invertible Networks*, SciPost Phys. **10** (2021) 6, 126, arXiv:2012.09873 [hep-ph].

[68] A. Butter, T. Heimel, T. Martini, S. Peitzsch, and T. Plehn, *Two invertible networks for the matrix element method*, SciPost Phys. **15** (2023) 3, 094, arXiv:2210.00019 [hep-ph].

[69] T. Heimel, N. Huetsch, R. Winterhalder, T. Plehn, and A. Butter, *Precision-Machine Learning for the Matrix Element Method*, arXiv:2310.07752 [hep-ph].

[70] R. K. Ellis, W. J. Stirling, and B. R. Webber, *QCD and collider physics*, vol. 8. Cambridge University Press, 2, 2011.

[71] T. Plehn, *Lectures on LHC Physics*, Lect. Notes Phys. **844** (2012) 1, arXiv:0910.4182 [hep-ph].

[72] J. Campbell, J. Huston, and F. Krauss, *The Black Book of Quantum Chromodynamics : a Primer for the LHC Era*. Oxford University Press, 2018.

[73] M. van Beekveld, S. Ferrario Ravasio, K. Hamilton, G. P. Salam, A. Soto-Ontoso, G. Soyez, and R. Verheyen, *PanScales showers for hadron collisions: all-order validation*, JHEP **11** (2022) 020, arXiv:2207.09467 [hep-ph].

[74] A. Andreassen, I. Feige, C. Frye, and M. D. Schwartz, *JUNIPR: a Framework for Unsupervised Machine Learning in Particle Physics*, Eur. Phys. J. C **79** (2019) 2, 102, arXiv:1804.09720 [hep-ph].

[75] A. Andreassen, I. Feige, C. Frye, and M. D. Schwartz, *Binary JUNIPR: an interpretable probabilistic model for discrimination*, Phys. Rev. Lett. **123** (2019) 18, 182001, arXiv:1906.10137 [hep-ph].

[76] T. Finke, M. Krämer, A. Mück, and J. Tönshoff, *Learning the language of QCD jets with transformers*, JHEP **06** (2023) 184, arXiv:2303.07364 [hep-ph].

[77] J. Bellm *et al.*, *Herwig 7.0/Herwig++ 3.0 release note*, Eur. Phys. J. C **76** (2016) 4, 196, arXiv:1512.01178 [hep-ph].

[78] T. Sjöstrand, S. Ask, J. R. Christiansen, R. Corke, N. Desai, P. Ilten, S. Mrenna, S. Prestel, C. O. Rasmussen, and P. Z. Skands, *An introduction to PYTHIA 8.2*, Comput. Phys. Commun. **191** (2015) 159, arXiv:1410.3012 [hep-ph].

[79] J. Alwall, R. Frederix, S. Frixione, V. Hirschi, F. Maltoni, O. Mattelaer, H. S. Shao, T. Stelzer, P. Torrielli, and M. Zaro, *The automated computation of tree-level and next-to-leading order differential cross sections, and their matching to parton shower simulations*, JHEP **07** (2014) 079, arXiv:1405.0301 [hep-ph].

[80] Sherpa, E. Bothmann *et al.*, *Event generation with Sherpa 3*, arXiv:2410.22148 [hep-ph].

[81] S. Catani, F. Krauss, R. Kuhn, and B. R. Webber, *QCD matrix elements + parton showers*, JHEP **11** (2001) 063, arXiv:hep-ph/0109231.

[82] M. L. Mangano, M. Moretti, F. Piccinini, R. Pittau, and A. D. Polosa, *ALPGEN, a generator for hard multiparton processes in hadronic collisions*, JHEP **07** (2003) 001, arXiv:hep-ph/0206293.

[83] S. Frixione, P. Nason, and C. Oleari, *Matching NLO QCD computations with Parton Shower simulations: the POWHEG method*, JHEP **11** (2007) 070, arXiv:0709.2092 [hep-ph].

[84] E. Bothmann, T. Childers, C. Gütschow, S. Höche, P. Hovland, J. Isaacson, M. Knobbe, and R. Latham, *Efficient precision simulation of processes with many-jet final states at the LHC*, Phys. Rev. D **109** (2024) 1, 014013, arXiv:2309.13154 [hep-ph].

[85] S. Höche, F. Krauss, and D. Reichelt, *The Alaric parton shower for hadron colliders*, arXiv:2404.14360 [hep-ph].

[86] M. R. Buckley, T. Plehn, and M. J. Ramsey-Musolf, *Top squark with mass close to the top quark*, Phys. Rev. D **90** (2014) 1, 014046, arXiv:1403.2726 [hep-ph].

[87] M. van Beekveld *et al.*, *Introduction to the PanScales framework, version 0.1*, SciPost Phys. Codeb. **2024** (2024) 31, arXiv:2312.13275 [hep-ph].

[88] M. van Beekveld *et al.*, *A new standard for the logarithmic accuracy of parton showers*, arXiv:2406.02661 [hep-ph].

[89] S. D. Ellis, R. Kleiss, and W. J. Stirling, *W's, Z's and Jets*, Phys. Lett. B **154** (1985) 435.

[90] F. A. Berends, W. T. Giele, H. Kuijf, R. Kleiss, and W. J. Stirling, *Multi - Jet Production in W, Z Events at $p\bar{p}$ Colliders*, Phys. Lett. B **224** (1989) 237.

[91] F. A. Berends, H. Kuijf, B. Tausk, and W. T. Giele, *On the production of a W and jets at hadron colliders*, Nucl. Phys. B **357** (1991) 32.

[92] E. Gerwick, T. Plehn, and S. Schumann, *Understanding Jet Scaling and Jet Vetos in Higgs Searches*, Phys. Rev. Lett. **108** (2012) 032003, arXiv:1108.3335 [hep-ph].

[93] E. Gerwick, T. Plehn, S. Schumann, and P. Schichtel, *Scaling Patterns for QCD Jets*, JHEP **10** (2012) 162, arXiv:1208.3676 [hep-ph].

[94] J. Alwall, M. Herquet, F. Maltoni, O. Mattelaer, and T. Stelzer, *MadGraph 5 : Going Beyond*, JHEP **06** (2011) 128, arXiv:1106.0522 [hep-ph].

[95] M. Cacciari, G. P. Salam, and G. Soyez, *FastJet User Manual*, Eur. Phys. J. C **72** (2012) 1896, arXiv:1111.6097 [hep-ph].

[96] M. Cacciari, G. P. Salam, and G. Soyez, *The anti-$k_t$ jet clustering algorithm*, JHEP **04** (2008) 063, arXiv:0802.1189 [hep-ph].

[97] J. Ackerschott, R. K. Barman, D. Gonçalves, T. Heimel, and T. Plehn, *Returning CP-observables to the frames they belong*, SciPost Phys. **17** (2024) 1, 001, arXiv:2308.00027 [hep-ph].

[98] T. Golling, L. Heinrich, M. Kagan, S. Klein, M. Leigh, M. Osadchy, and J. A. Raine, *Masked particle modeling on sets: towards self-supervised high energy physics foundation models*, Mach. Learn. Sci. Tech. **5** (2024) 3, 035074, arXiv:2401.13537 [hep-ph].

[99] M. Leigh, S. Klein, F. Charton, T. Golling, L. Heinrich, M. Kagan, I. Ochoa, and M. Osadchy, *Is Tokenization Needed for Masked Particle Modelling?*, arXiv:2409.12589 [hep-ph].

[100] J. Birk, A. Hallin, and G. Kasieczka, *OmniJet-α: the first cross-task foundation model for particle physics*, Mach. Learn. Sci. Tech. **5** (2024) 3, 035031, arXiv:2403.05618 [hep-ph].

[101] D. MacKay, *Probable Networks and Plausible Predictions – A Review of Practical Bayesian Methods for Supervised Neural Networks*, Comp. in Neural Systems **6** (1995) 4679.

[102] R. M. Neal, *Bayesian learning for neural networks*. PhD thesis, Toronto, 1995. ftp://www.cs.toronto.edu/dist/radford/thesis.pdf.

[103] Y. Gal, *Uncertainty in Deep Learning*. PhD thesis, Cambridge, 2016. http://mlg.eng.cam.ac.uk/yarin/thesis/thesis.pdf.

[104] A. Kendall and Y. Gal, *What Uncertainties Do We Need in Bayesian Deep Learning for Computer Vision?*, Proc. NIPS (2017) , arXiv:1703.04977 [cs.CV].

[105] G. Kasieczka, M. Luchmann, F. Otterpohl, and T. Plehn, *Per-Object Systematics using Deep-Learned Calibration*, SciPost Phys. **9** (2020) 089, arXiv:2003.11099 [hep-ph].

[106] ATLAS, G. Aad *et al.*, *Precision calibration of calorimeter signals in the ATLAS experiment using an uncertainty-aware neural network*, arXiv:2412.04370 [hep-ex].

[107] S. Bollweg, M. Haußmann, G. Kasieczka, M. Luchmann, T. Plehn, and J. Thompson, *Deep-Learning Jets with Uncertainties and More*, SciPost Phys. **8** (2020) 1, 006, arXiv:1904.10004 [hep-ph].

[108] M. Bellagente, M. Haussmann, M. Luchmann, and T. Plehn, *Understanding Event-Generation Networks via Uncertainties*, SciPost Phys. **13** (2022) 1, 003, arXiv:2104.04543 [hep-ph].

[109] S. Bieringer, S. Diefenbacher, G. Kasieczka, and M. Trabs, *Calibrating Bayesian generative machine learning for Bayesiamplification*, Mach. Learn. Sci. Tech. **5** (2024) 4, 045044, arXiv:2408.00838 [cs.LG].

[110] M. Backes, A. Butter, T. Plehn, and R. Winterhalder, *How to GAN Event Unweighting*, SciPost Phys. **10** (12, 2021) 089, arXiv:2012.07873 [hep-ph].

[111] S. Rizvi, M. Pettee, and B. Nachman, *Learning Likelihood Ratios with Neural Network Classifiers*, arXiv:2305.10500 [hep-ph].