# Peer review of "Extrapolating Jet Radiation with Autoregressive Transformers"

_SciPost Physics_

## Round 1 · Referee Report · Anonymous (Referee 1) · 2025-12-3

Disclosure of Generative AI use

The referee discloses that the following generative AI tools have been used in the preparation of this report:

I used Gemini 3 Pro to check whether my comments were satisfactorily addressed in the PDF diff

Report

This is an innovative study of how transformer-based jet radiation models can potentially extrapolate to larger numbers of jets, shedding light on architectural choices and training techniques that facilitate this capability. I'd like to thank the authors for thoroughly considering and implementing the comments from my first report. The text of the new manuscript is clearer / better nuanced on several key points and the figures are enhanced. The addition of a diagram of the training workflow (Figure 7) is especially helpful. I do not have further comments and am happy to recommend publishing this work.

Recommendation

Publish (easily meets expectations and criteria for this Journal; among top 50%)

---

## Round 1 · Author Response

Dear Editor and Referees,

We thank the referees for their time, careful consideration, and the evaluation of our manuscript. We list below the changes we have made concerning the helpful suggestions.

Report #1:

1) It would enhance the paper’s appeal to a broader audience to state how this generative model, in particular, could benefit future collider experiments. If the model can accurately generate Z+jets events, could it do so with better alignment to real data? You mention new analysis techniques – could you expand on that? Traditional event generators are already pretty fast, so speed is probably not the main opportunity.

We included a few lines in the introduction making references to these points.

2) The linear binning in Figures 6, 7, 9, and 11 makes it difficult to judge the agreement with truth beyond the bulk of the distribution, which is critical for momentum spectra. This is especially true for the sum-of-pT distributions where the x-axis range seems a bit prematurely cut off. Could you please try logarithmic binning or even simply increasing the bin width? Similarly, the ratio panels would probably become more informative with smaller statistical uncertainties on the truth and perhaps a larger y-axis range.

We have improved the readability of the Figures by 1. using wider bins and adjusting the x-axis range case-by-case 2. increasing the y-axis range in the ratio panels We found that a logarithmic binning was not helpful because the distributions do not span many orders of magnitude in momentum.

3) For each of the extrapolation techniques, after pointing out the areas of clear mismodelling, the authors argue that these results demonstrate that “autoregressive transformers can learn the universal nature of jet radiation.” This appears to beg the question: “why is there mismodelling if the model in fact learned universal jet behavior”? Either the variables considered do not actually test these universal properties, or, if they do, the model failed to learn them (at least beyond the training data).

Given a limited amount of training data and training time, a fully expressive neural network will not learn a certain behaviour to perfect precision, even if there is exact universality. However, we note that the naive approach clearly fails to extrapolate, whereas our approaches with modified losses lead to distributions that agree with the unseen truth to a precision covered by the learned uncertainties. We therefore believe that our findings support the claims in the results section.

4) This paper includes an appendix of nearly 6 pages discussing a novel approach that is otherwise only mentioned in the last sentence of the outlook section. How could this side-investigation be more naturally included in the paper’s main discussion? If this proves too difficult, have you considered publishing the DiscFormer work in a separate paper?

We thank the referee for this comment. The DiscFormer approach indeed represents a more exploratory side-investigation within our study. While the method works in certain cases, our comparison with a standard generator shows that it performs marginally better, and it is not yet clear to us whether it can ultimately provide a substantial benefit for collider simulations, given the additional overhead. For this reason, we chose to include the results as an extended appendix, where they can serve primarily as a demonstration of the idea and the performance we observe, rather than as a central message of the paper. We agree that a more prominent discussion would require stronger evidence of impact, which at this stage we do not yet have.

5) From what I can tell, you don’t generate jet constituents, but rather the jets themselves. How hard would it be to couple your jet-level transformer with a constituent-level transformer? The latter could be conditioned on the jet features and could be evaluated independently for each jet. This limitation/opportunity might be worth discussing in the Outlook section, but it’s totally up to you.

We agree that the approach can be extended to jet constituent generation, and we are investigating this direction. We added a sentence to the outlook.

6) Transverse momentum conservation is a condition you check to assess model performance, but what if you could incorporate it directly into the model design? This way, the model would not have to learn it implicitly and might converge better. It appears that you actually do something similar in your “override” technique when you use momentum conservation to help guide the model’s extrapolation to high jet multiplicity.

Indeed, we could implement exact pT conservation in generated events (for example by not sampling the px and py of the last particle but calculating it instead). However, in our case pT is not exactly conserved due to detector effects. We want to reproduce the approximate conservation, but this is difficult to do in general.

7) Abstract: “Autoregressive transformers allow us to generate events with variable numbers of particles, very much in line with the physics of QCD jet radiation.” ⇒ This makes it sound like autoregressive transformers are unique in this respect, but there are other ways to handle variable-length outputs. I would suggest: “Autoregressive transformers are an effective approach to generate events…”

We agree that our original wording was not very precise. The unique aspect of an autoregressive model is the possibility to extrapolate. We reworded the sentence in the abstract to include this.

8) P2: “The goal of this paper is to show, for the first time, that a generative transformer can extrapolate in the number of jets and generate approximately universal jet radiation for higher jet numbers than seen during the training.” ⇒ Can the models in the above references also extrapolate, or is this unique to the autoregressive architecture?

Ref. 27 (‘Jet Diffusion versus JetGPT’) focuses on generating events with fixed, predetermined multiplicity. The network is trained on a fixed range of multiplicities, and when generating events the multiplicity is always predetermined. Each jet receives a different one-hot encoding, preventing extrapolation. Refs 72+73 (‘JUNIPR’) and Ref 74 (‘Language of QCD Jets’) both use autoregressive approaches which can in principle be used for extrapolation. However, Refs 72+73 do not study extrapolation, and their approach can also not be used to generate events. Ref 74 covers extrapolation, but only in a partial way as they include the leading 50 constituents of jets with more than 50 constituents in the training. Additionally, these two directions both discuss the structure of jet constituents, whereas we focus on structures between jets.

9) P3: “The iterative structure of Eq.(1) allows us to simulate parton splittings as Markov processes” ⇒ If the splittings are strictly Markovian, doesn’t that imply that the probability for each to occur is agnostic to the details of the preceding splitting history? If this assumption is valid, why do we need a transformer to model the sequence of splittings? There would be nothing to learn about the correlations between one splitting and the next, which are exactly what attention is designed to capture. In this case, it wouldn’t be incorrect to use a transformer, but it should be equally effective to use a simpler model that ignores correlations between elements in the set. It would be instructive to clarify this point.

We agree with the referee that no transformer is needed if the Markovian structure was perfectly preserved in the sequence of jets that we are trying to describe with a generative network. However, the sentence “The iterative structure of Eq.(1) allows us to simulate parton splittings as Markov processes” appears within the parton shower introduction. At the level of the parton shower, the structure is indeed Markovian, up to corrections outlined in the next paragraph. The jet algorithm then clearly breaks this Markovian structure. However, jet radiation still features scaling patterns, which are remainders of the Markovian structure of the parton shower. The symmetry breaking effects discussed above induce non-Markovian correlations in jet radiation, which require a transformer to be modelled accurately. We added a clarifying sentence when motivating the autoregressive factorization in Section 2.3.

10) P4: “2.2 Z + jets dataset” ⇒ It appears that MadGraph, Pythia, anti-kT, and FastJet should to be cited. Similarly, it would be better to either cite, unpack, or drop the reference to “CKKW”.

Thank you, we added the references.

11) P5: “500M events” ⇒ Please state the CoM energy and whether these are proton-proton collisions

We extended the text to say that we generate pp events at a center-of-mass energy of 13TeV.

12) P5: “The jets are defined with FASTJETv3.3.4 using the anti-kT algorithm” ⇒ R0.4 jets? Any rapidity cuts?

We use the default R=0.4 and do not apply any rapidity cuts.

13) P5: “muons” ⇒ are there also Z->ee events or only muons? Just a clarification, since only “leptonically” was specified above.

You are right, we changed to “decaying to muons”.

14) P5: “ordered in transverse momentum” ⇒ ordered in descending transverse momentum (?)

Thank you, we fixed the sentence based on your suggestion.

15) P5: “For 10 jets the phase space is 45-dimensional.” ⇒ Shouldn’t it be 410 + 32 = 46? Is the leading muon phi degree of freedom squashed?

We indeed squash the leading muon phi due to the global rotation symmetry. We added a clarifying sentence.

16) P6: “the sequence of particles x1, . . . , xi−1” are you referring here to jet constituent particles following Parton shower (many) or the progenitor partons (few) that correspond roughly to individual jets?

We changed particles to progenitor partons and added later a sentence saying that “we also include the muons in the sequence”.

17) P6: “The Kronecker delta” ⇒ Add “\delta_in”

We added the 𝛿 as suggested.

18) P7: “For Zµµ+jets events, we also treat the muons autoregressively and enforce a splitting probability of one for them.” ⇒ Why would you have the model learn this when it is obvious from physics considerations that the muons must be present? You could simply focus on the hadronic content of the event.

We included the muons to capture the full event, although as you say they are not necessary to study the jet kinematics extrapolation. We reworded the sentence to “$Z_{\mu\mu}+$jets events, we also include the muons in the sequence, but explicitly set their splitting probabilities to one instead of learning them.” to make this clearer.

19) P7: “we factorize the likelihood of individual particles pkin(xi+1|x1:i) in terms of their components. The ordering of components can affect the network performance” ⇒ This makes it sound like each particle’s pt, eta, phi, mass get generated one after the other as a sequence (just like the particles themselves). I think that all you mean to say is that you split the joint distribution into a product of 4 separate 1-D distributions. However, looking closely at Eq. 17, I see that each subsequent factor also uses as input the sampled kinematic value from the previous factor. This was not immediately clear to me. Can you please try to clarify the text, and also motivate why these “i+1” dependencies are useful?

Indeed the pt, eta, phi and mass of each particle are generated one-by-one as a sequence. Keeping the conditioning on the previous components (the i+1 dependencies) is necessary to ensure that the decomposition is completely general — without them we would lose correlations. The purpose of the decomposition is to allow easy sampling and likelihood evaluation, since we end up with a product of Gaussian mixtures. We updated the wording of the key sentence to “Similarly to the decomposition of the event likelihood p(x_{1:n}), we autoregressively factorize the likelihood of individual particles” and added text to clarify that this makes sampling/likelihood easy.

20) P8: “Autoregressive transformer” ⇒ There are a few details missing about inputs and outputs. Please update the text to address the following questions: a) Input: - What is the start token? - Which event-level features are used to condition the generation, and how is it embedded?

Since we generate the events unconditionally, the start token contains only zeros. It is only included so that each network call has the same signature, which simplifies the implementation.

21) b) Output: - Are generated particles ordered in decreasing pT? I.e. is the last jet generated always the lowest-pT? - Do the muons have to be generated from scratch or do you copy the from the generator? - Is particle ID generated, or is there some other way to distinguish the muons?

We do not enforce that jets are generated ordered by decreasing pT. Instead, we manually order the jets after the autoregressive generation process by decreasing pT, to match the pT-ordering in the test dataset. It is possible to generate particles already ordered by pT, for instance by generating the particle pT’s as a fraction of the previous particles pT and constraining this fraction to be smaller than 1. We have experimented with this approach, but found worse performance. The muons are generated from scratch. To specify the ID of the particle that should be generated next, we give a one-hot encoding of the next particle’s particle ID as a condition to the network. There are three possible values, 0 and 1 to specify that the next particle should be one of the incoming muons, and 2 if it should be a jet.

22) P13: “However, there are deviations in the kinematic features from the truth that are not covered by the Bayesian uncertainty.” ⇒ This makes sense. It seems that the bootstrapping procedure would bring the 7-jet events into the training distribution in terms of their cardinality, but not quite in terms of their (potentially different) kinematics, since this wouldn’t have been learned properly by the network.

We added a clarifying sentence.

23) P9: “The jet multiplicity distribution is shown in Fig. 4” ⇒ Fig. 4 mentioned before Fig. 3

We have exchanged the order of the two figures.

24) P14: “3.4 Extrapolation with override” ⇒ Should reference Fig. 8 again here (right-hand side)

The figure is briefly mentioned “Similarly to the truncated loss, this override approach significantly increases the fraction of higher-multiplicity”. However, it was disconnected from the reference to the Figure. We restructured the discussion in Section 3.4 to improve readability and extended the sentence with “...compared to the naive extrapolation” to emphasize the message to be taken from the Figure.

25) P3: “might be not be sufficient” P16: “this override approach significantly increasing the fraction” P17: “We find that all both approaches” ⇒ I assume you meant to say that “truncated” and “override” both perform similarly well, improving over naive extrapolation and bootstrapping.

We fixed these typos.

Report #2:

1) Regarding point 1 in "weaknesses": A figure visualizing the bootstrapping workflow, combined with a more explicit step-by-step explanation in the main text, would significantly improve the clarity of this method.

We created a figure to clarify the bootstrapping workflow. We think that together with the second paragraph in Section 3.2 the bootstrapping workflow is described in sufficient detail.

2) If possible, please provide some possible explanations for the differences in the generated kinematic distributions when using the two different losses: truncated vs override.

We added a comment regarding the improvement in momentum conservation for the override method “This is to be expected, since the override loss is specifically designed to match the global momentum distribution.”

---

## Round 1 · List of Changes

Changes in the text are marked in blue and red in the attached PDF. Besides changes in the text, we added Figure 7 and modified the plotting style in Figures 4, 6, 8, 10, 12.

---

## Editorial Decision

accepted_in_target_journal